# The appendicular myology of *Stegoceras validum* (Ornithischia: Pachycephalosauridae) and implications for the head-butting hypothesis

Bryan R. S. Moore[ID]^1*, Mathew J. Roloson^1, Philip J. Currie[ID]^2, Michael J. Ryan^1,3, R. Timothy Patterson^1, Jordan C. Mallon^1,3

**1** Ottawa Carleton Geoscience Center and Department of Earth Sciences, Carleton University, Ottawa, Ontario, Canada, **2** Department of Biological Sciences, University of Alberta, Edmonton, Alberta, Canada, **3** Beaty Centre for Species Discovery and Palaeobiology section, Canadian Museum of Nature, Ottawa, Ontario, Canada

* bryanmoore@cmail.carleton.ca

## Abstract

In this study, we use an exceptional skeleton of the pachycephalosaur *Stegoceras validum* (UALVP 2) to inform a comprehensive appendicular muscle reconstruction of the animal, with the goal of better understanding the functional morphology of the pachycephalosaur postcranial skeleton. We find that *S. validum* possessed a conservative forelimb musculature, particularly in comparison to early saurischian bipeds. By contrast, the pelvic and hind limb musculature are more derived, reflecting peculiarities of the underlying skeletal anatomy. The iliotibialis, ischiocaudalis, and caudofemoralis muscles have enlarged attachment sites and the caudofemoralis has greater leverage owing to the distal displacement of the fourth trochanter along the femur. These larger muscles, in combination with the wide pelvis and stout hind limbs, produced a stronger, more stable pelvic structure that would have proved advantageous during hypothesized intraspecific head-butting contests. The pelvis may have been further stabilized by enlarged sacroiliac ligaments, which stemmed from the unique medial iliac flange of the pachycephalosaurs. Although the pubis of UALVP 2 is not preserved, the pubes of other pachycephalosaurs are highly reduced. The puboischiofemoralis musculature was likely also reduced accordingly, and compensated for by the aforementioned improved pelvic musculature.

## Introduction

Pachycephalosauria is a group of generally small, bipedal ornithischian dinosaurs, known from the Santonian to Maastrichtian (~86 to 66 Ma) of Asia and North America [1–3]. These animals are characterized by frontal and parietal bones (and sometimes associated bones such as the squamosals and postorbitals) that are fused into a thickened skull dome. The dome is often adorned around its lateral and posterior margins by rounded nodes or spikes [2, 4, 5].

**Data Availability Statement:** All relevant data are within the paper and its Supporting Information files.

**Funding:** This study is supported by a Natural Sciences and Engineering Research Council of Canada (https://www.nserc-crsng.gc.ca/) Discovery Grant to JCM (RGPIN-2017-06356). This study is also supported by a Dinosaur Research Institute (https://www.dinosaurresearch.com) Student Project Grant to BRSM. Additionally, this study is supported by the Dale Patten Memorial Fund acquired by JCM. The funders had no role in study design, data collection and analysis, decision to publish, or preparation of the manuscript.

**Competing interests:** The authors have declared that no competing interests exist.

**Abbreviations:** *Institutional abbreviations:* CMN, Canadian Museum of Nature, Ottawa, Ontario, Canada; MPC, Palaeontological Center, Mongolian Academy of Sciences, Ulaanbaatar, Mongolia; ROM, Royal Ontario Museum, Toronto, Ontario, Canada; UALVP, University of Alberta Laboratory of Vertebrate Paleontology, Edmonton, Alberta, Canada.

Because of their robust nature, the domes preserve much more readily in the fossil record compared to the delicate postcranial skeleton [6, 7]. As a result, much research about these animals has concentrated on the form and function of the dome, which has been implicated in intraspecific head-butting contests (either head-on-head or head-on-flank butting) [8–15].

In spite of the relative lack of preserved postcrania, the few pachycephalosaur skeletons that are preserved show a unique morphology. The hind limbs are remarkably stout compared to other small, bipedal ornithischians (e.g., *Hypsilophodon foxii*, *Lesothosaurus diagnosticus*, *Thescelosaurus* spp.), and the torso and pelvic region are comparably broad [2]. The pelvic girdle displays a medially projecting tab from the dorsal margin of the ilium and a highly reduced pubis that is almost entirely excluded from the acetabulum [2]. The pre- and postzygapophyses of the dorsal vertebrae articulate in double ridge-and-groove structures that provide rigidity to the spine. The base of the tail is also wide and surrounded by a caudal basket of myorhabdoid ossifications [16].

In this study, we use an exceptional specimen of the pachycephalosaur *Stegoceras validum* (UALVP 2), which preserves a nearly complete appendicular skeleton [17], to reconstruct the appendicular myology of a pachycephalosaur for the first time. We also provide a detailed consideration of the functioning of the postcranium for use in purported head-butting contests [13, 18].

## Materials and methods

Muscles generally do not preserve in the fossil record, making it exceedingly difficult to understand the myology of fossil organisms. Fortunately, direct evidence of muscles is often preserved by the surface texture of bones. Osteological correlates of muscles develop as muscles exert stress on the underlying bones [19–22]. Such textures indicate where muscles would have attached to the bone, the minimum size of the attachment area, and even the type of attachment. Unfortunately, not all muscles will produce visible texturing, nor will the entire attachment area be preserved [19, 21–24]. As such, when reconstructing the myology of fossil organisms, it is often necessary to rely on comparisons with the Extant Phylogenetic Bracket (EPB) to constrain the possible range of interpretations [25, 26]. The EPB for non-avian dinosaurs includes birds and crocodilians. By observing the myology of dinosaur relatives, one can infer where and how muscles may have attached in fossil organisms [26, 27]. Of course, this method of reconstruction is not infallible, as extant organisms often have different body plans than their fossil predecessors and have likely modified their myology as a result [26, 27]. Nevertheless, the EPB provides a good null model by which to reconstruct soft tissues in extinct taxa.

Here, we reconstruct the appendicular myology of *S. validum* using UALVP 2. This specimen includes one of the best preserved pachycephalosaur postcranial skeletons discovered to date and the best preserved pachycephalosaur from Canada. In addition to having a wide variety of preserved postcranial elements, the quality of preservation is also exceptional. Surface textures are clearly visible on almost all elements making the specimen ideal to examine for osteological correlates of soft tissue structures. There is also minimal postdepositional deformation of postcranial elements and those that are deformed may be supplemented by their left/right counterparts.

To reconstruct the appendicular musculature of UALVP 2, high-fidelity plaster casts of UALVP 2 were first analyzed alongside photographs and 3D scans of the original material to identify any osteological correlates (striations, pitted/rugose textures, raised structures, etc.). These observations were then compared to the musculature of crocodilians and birds [28–31] to verify their agreement. Muscle insertion and origin sites were drawn on bone outlines in

Inkscape version 1.0 (2020) [32] and 3D muscle reconstructions were created using Blender version 2.7 (2018) [33]. Because muscles do not always create surface texturing, and when they do it is usually not over their entire attachment sites [19, 24], muscles for UALVP 2 were only reconstructed over the area of their corresponding osteological correlates. This leads to a highly conservative reconstruction with a muscle distribution pattern that is likely sparser than it was in life. For muscles that have multiple attachment sites which are not all preserved in UALVP 2, the missing attachments were inferred using EPB comparisons. Muscle terminology follows standard veterinary anatomy.

When reconstructing the myology of extinct animals, there is always a degree of uncertainty. To aid in understanding the degree to which such reconstructions can be trusted, Witmer (1995) [26] identified several levels of inference. These will be used here when describing the musculature of *S. validum*. Level 1 inferences are those with the highest degree of certainty in which both the soft tissue and the osteological correlate(s) are present in the EPB. Level 1' inferences are those where the soft tissue is present in the EPB, but the osteological correlate is not or it differs between taxa. Less confidence is given to cases where the soft tissue and osteological correlate(s) are present in only some of the EPB taxa: level 2 inferences.

## Results

The appendicular skeleton of UALVP 2 preserves elements of the pectoral girdle, pelvic girdle, forelimb, and hind limb. These elements include the right scapula, right coracoid, right humerus, right radius, right ulna, right ilium, right ischium, left and right femora, left tibia, left and right fibulae, and right metatarsals I-IV. The following reconstruction will describe the muscles that attached to each of these elements and infer muscle attachments for elements of the appendicular skeleton that were not preserved, based on comparisons with the Extant Phylogenetic Bracket. Reconstructed muscle origins and insertions for *S. validum* are listed in Tables 1 and 2.

### Pectoral and brachial musculature

**Levator scapulae (LS).**   The presence of the levator scapulae is debated in dinosaurs [21, 23, 34] because it is present in modern crocodilians but not in birds [31, 35–37]. It has been reconstructed in dinosaurs such as *Dreadnoughtus schrani* Lacovara *et al.*, 2014 [21], *Opisthocoelicaudia skarzynskii* Borsuk-Bialynicka, 1977 [38], and *Tawa hallae* Nesbitt *et al.*, 2009 [35], but has been left out in others, such as dromaeosaurids [37]. In crocodilians and the aforementioned dinosaurs for which this muscle has been reconstructed, its superficial insertion is always found along the anterior/dorsal edge (depending on the orientation of the bone in the animal) of the scapular blade with a corresponding osteological correlate [21, 31, 35–38]. Based on the presence of striations in this area in UALVP 2, the insertion site of the muscle in crocodilians, and its reconstruction in numerous groups of non-avian dinosaurs, the LS is reconstructed in *S. validum* as inserting along the distal third and dorsal edge of the scapular blade (Fig 1). The origin of the LS is equivocal in dinosaurs but would likely have been on the anterior cervical ribs as it is in crocodilians [31, 35]. This muscle rotated the scapular blade and laterally flexed the neck [35]. If present, the trapezius would have inserted with the LS; however, the trapezius has no osteological origin in archosaurs [21, 35, 37]. Instead, it originates from the underlying cervico-thoracic musculature. Without an osteological origin, the trapezius is not reconstructed here in *S. validum*.

**Rhomboideus (RH).**   The presence of the rhomboideus is unequivocal in dinosaurs; however, its number of divisions is equivocal [21, 23, 31, 34, 35, 37]. A profundus division is only found in birds and is reconstructed in dromaeosaurids [35, 37], based on the sub-horizontal

**Table 1. Reconstructed origins and insertions of the pectoral and forelimb musculature of *Stegoceras validum*.** Inference levels are given in parentheses in second and third columns.

| Muscle | Origin | Insertion |
|---|---|---|
| Levator scapulae (LS) | Anterior cervical ribs (2) | Dorsal edge of distal scapula (2) |
| Rhomboideus (RH) | Faschia of the cervico-thoracic region and neural spines of the posterior cervical and anterior dorsal vertebrae (1') | Distomedial scapula (1') |
| Serratus superficialis (SRS) | Lateral surfaces of the anterior dorsal ribs (1) | Posteroventral margin of the distal scapula (1') |
| Serratus profundus (SRP) | Lateral surfaces of the anterior dorsal ribs (1) | Posteroventral medial surface of the distal scapula (1) |
| Deltoideus scapularis (DSC) | Lateral surface of the distal scapula (1') | Lateral surface of the deltopectoral crest (1) |
| Deltoideus clavicularis (DCL) | Acromial ridge of the scapula (1) | Lateral surface of the deltopectoral crest, distal to the DSC |
| Subscapularis (SBS) | Medial scapular blade proximal to the RH and SRP (1) | Proximomedial surface of the humerus (1) |
| Subcoracoideus (SBC) | Medial coracoid surrounding the coracoid foramen (2) | Shared with the SBS (1) |
| Scapulohumeralis posterior (SHP) | Lateral scapular blade (1') | Posterior humerus distal to the internal tuberosity (1) |
| Scapulohumeralis anterior (SHA) | Proximal posteroventral margin of the scapular blade (2) | Posterior humerus distolateral to the internal tuberosity (2) |
| Supracoracoideus longus (SCL) | Scapula along the coracoid suture (2) | Proximolateral deltopectoral crest (1') |
| Supracoracoideus brevis (SCB) | Medial ventral margin of the coracoid (1') | Proximolateral deltopectoral crest (1') |
| Coracobrachialis (CB) | Posteroventral lateral surface of the coracoid (1) | Proximoanterior surface of humerus (1) |
| Triceps brachii (TB) | Proximoventral scapula (1) and posteromedial humeral shaft (1) | Olecranon process of the ulna (1) |
| Biceps brachii (BB) | Lateral coracoid anteroventral to the coracoid foramen and dorsal to the CB (1) | Posterolateral surface of the proximal radius (1) |
| Pectoralis (P) | N/A | Apex of the deltopectoral crest (1) |
| Brachialis (BR) | Anterolateral humeral shaft distal to the deltopectoral crest (1') | Shared with BB (1) |
| Latissimus dorsi (LD) | Neural spines of posterior cervical and anterior dorsal vertebrae (1) | Proximolateral portion of the posterior humeral shaft (1) |
| Anconeus (AN) | Distal humeral ectepicondyle (1') | Lateral ulnar shaft (1) |
| Extensor carpi ulnaris (ECU) | Distal humeral ectepicondyle (1') | N/A |
| Supinator (SU) | Proximal humeral ectepicondyle (1') | Anterolateral radius (1) |
| Extensor carpi radialis (ECR) | Humeral ectepicondyle between the SU and extensor digitorum longus (1') | Radiale (1') |
| Extensor digitorum longus (EDL) | Middle of the humeral ectepicondyle (1) | Base of metacarpals I-IV (1) |
| Abductor radialis (AR) | Humeral ectepicondyle lateral to the ECR (2) | Proximolateral radius (1') |
| Abductor pollicis longus (APL) | Lateral ulnar shaft and distomedial radial shaft (1) | Metacarpal I (1') |
| Pronator teres (PT) | Proximal entepicondyle (1) | Anteromedial radius (1') |
| Pronator quadratus (PQ) | Medial ulnar shaft (1') | Ulnar-facing radius (1') |
| Flexor carpi ulnaris (FCU) | Distal humeral entepicondyle (1') | Pisiform (1) |
| Flexor digitorum longus superficialis (FDLS) | Middle of humeral entepicondyle (1) | Ventral distal phalanges (1') |
| Flexor digitorum longus profundus (FDLP) | Medial ulnar shaft (1') | Shared with FDLS (1') |

position of the scapula. There are no osteological correlates directly supporting the presence of a profundus division of the RH in UALVP 2. A horizontal scapular position also results in an insertion of the RH superficialis along the medial side of the anterodorsal edge of the distal half of the scapular blade [35, 37]. In crocodilians and ancestral theropods, the scapula is oriented sub-vertically and possesses no profundus division [35, 39]. The more vertical scapular orientation also leads to a lower insertion of the RH on the medial surface of the distal half of the scapula [35, 39]. In pachycephalosaurs, the scapula would have been oriented intermediate between the avian and crocodilian conditions [1, 40–42]. As such, it is likely that the RH

**Table 2. Reconstructed origins and insertions of the pelvic and hind limb musculature of *Stegoceras validum*.** Inference levels are given in parentheses in second and third columns.

| Muscle | Origin | Insertion |
|---|---|---|
| Iliocaudalis (ILC) | Posterolateral postacetabular process of the ilium (1) | Transverse processes and haemal spines of the anterior caudal vertebrae (1) |
| Ischiocaudalis (ISC) | Distolateral ischial shaft (1) | Shared with the ILC (1) |
| Puboischiofemoralis internus 1 (PIFI 1) | Medial preacetabular process of the ilium (1) | Posteromedial femur medial to the caudofemoralis longus (1) |
| Puboischiofemoralis internus 2 (PIFI 2) | Lateral surfaces of dorsal vertebrae (1) | Lesser trochanter of femur (1) |
| Iliofemoralis (ILFE) | Supracetabular region of the lateral ilium (1') | Shared with PIFI 2 (1') |
| Puboischiofemoralis externus 3 (PIFE 3) | Lateral ischium (2) | Posterolateral greater trochanter of the femur (1) |
| Iliotibialis (ILT) | Dorsal margin of the ilium (1) | Anteromedial cnemial crest of the tibia (1) |
| Iliofibularis (ILF) | Dorsolateral margin of the postacetabular process of the ilium (1) | Anterolateral fibula (1) |
| Flexor tibialis internus 3 (FTI 3) | Lateral ischium just distal to the iliac peduncle (1') | Posteromedial tibia (1) |
| Flexor tibialis externus (FTE) | Lateral postacetabular process of the ilium posteroventral to the ILF (1) | Shared with the FTI 3 (1) |
| Caudofemoralis brevis (CFB) | Ventrolateral postacetabular process of the ilium (1) | Posterior femur lateral to the fourth trochanter (1) |
| Caudofemoralis longus (CFL) | Lateral caudal centra (1') | Posterior femur medial to the fourth trochanter (1) |
| Adductor femoralis 1 (ADF 1) | Dorsolateral ischial shaft (1') | Posterodistal femur between the lateral and medial condyles (1) |
| Adductor femoralis 2 (ADF 2) | Ventrolateral ischial shaft (1') | Shared with ADF 1 (1) |
| Ischiotrochantericus (ISTR) | Dorsomedial ischial shaft (1') | Posterolateral femur just distal to the PIFE complex (1') |
| Ambiens (AMB) | Proximolateral prepubis (2) | Shared with the ILT and femorotibiales (1) |
| Femorotibialis lateralis (FMTL) | Lateral surfaces of the anterior and posterior femoral shaft (1') | Shared with the ILT and AMB (1) |
| Femorotibialis medialis (FMTML) | Medial surfaces of the anterior and posterior femoral shaft (1') | Shared with the ILT and AMB (1) |
| Gastrocnemius lateralis (GSCL) | Posteromedial lateral condyle of the femur (1) | Ventral surfaces of metatarsals II-IV (1) |
| Gastrocnemius medialis (GSCM) | Posteromedial medial condyle of the femur (1) | Shared with the GSCL (1) |
| Fibularis longus (FBL) | Lateral fibular shaft posterior to the FBB origin (1') | Ventral calcaneum and distoventral metatarsal V (1) |
| Fibularis brevis (FBB) | Lateral fibular shaft anterior to the FBL origin (1') | Ventral calcaneum and distoventral metatarsal V (1) |
| Extensor digitorum longus (EDL) | Lateral surface of the lateral distal condyle of the femur or anterior cnemial crest (1') | Dorsal unguals and phalanges (1') |
| Extensor digitorum brevis (EDB) | Dorsal surface of the proximal tarsals (1') | Shared with EDL (1') |
| Flexor digitorum longus (FDL) | Posterolateral surface of the lateral femoral condyle and posterolateral proximal fibula (1') | Ventral unguals and phalanges II-IV (1) |
| Popliteus (POP) | Posterior cnemial pocket of the tibia (1') | Proximomedial fibula (1) |
| Tibialis anterior (TBA) | Anterior cnemial pocket of the tibia (1') | Proximolateral metatarsals II-IV (1) |
| Interosseus cruris (IOC) | Distal anterolateral tibial shaft (1) | Distomedial fibula (1) |
| Pronator profundus (PP) | Distal posterolateral tibia (1) and distal posteromedial fibula (2) | Ventromedial metatarsal II (1) |

superficialis would have inserted at an intermediate position between the dorsal margin and medial surface of the scapular blade similar to that proposed for *T. hallae* [35]. This is supported by the presence of parallel striations in this location on the scapula of UALVP 2. With an intermediate scapular orientation, it is likely that the origin of the RH was also intermediate between the conditions of crocodilians and birds, attaching to the fascia and neural spines of the posterior cervical and anterior dorsal vertebrae [35]. Based on the proposed orientation of the scapula and morphology of the RH, this muscle protracted the scapula [35].

**Serratus (SR).** The presence of the serratus musculature is unequivocal in dinosaurs [21, 23, 34, 35, 37]. It originates from the lateral surfaces of the anterior dorsal ribs in both birds and crocodilians [35, 37]. There is no reason to think this was not the case in *S. validum*. The

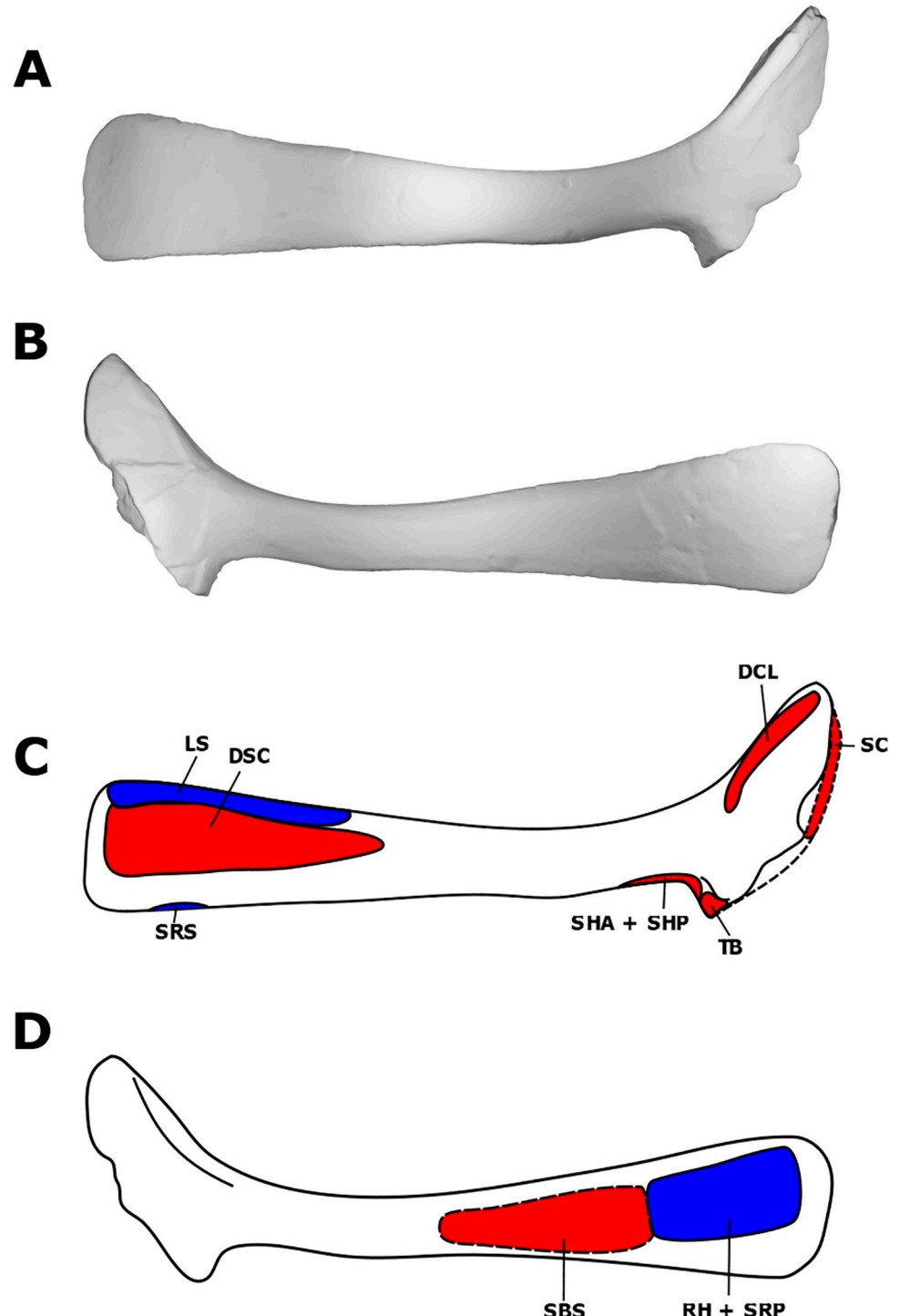

**Fig 1. Myological reconstruction and scans of the right scapula of *Stegoceras validum*.** (A) Lateral scan, (B) medial scan, (C) lateral reconstruction, and (D) medial reconstruction. Areas of red indicate muscle origins. Areas of blue indicate muscle insertions. Dashed lines indicate muscles that were reconstructed entirely based on EPB comparisons.

insertion point(s) of the serratus superficialis (SRS) in dinosaurs are equivocal [21, 35, 37]. Jasinoski *et al.* (2006) proposed two points of insertion for dromaeosaurids, based on a tubercle present in *Ingenia yanshini* (Barsbold, 1981) and neognath birds. More conservative

models, such as those for *T. hallae* [35] and *D. schrani* [21], propose a single elongated attachment site along the posteroventral margin of the distal half of the scapula. In UALVP 2, the only osteological correlate in this area is a small rugosity at the distal end of the posteroventral margin of the scapular blade. As such, the insertion of the SRS is conservatively reconstructed as being limited to this correlate (Fig 1).

The serratus profundus (SRP) inserts on the medial surface of the distal end of the scapular blade in both birds and crocodilians [35, 37]. Similar insertions have been proposed for *T. hallae* [35] and *D. schrani* [21]. The same area displays a striated texture on the scapula of UALVP 2, part of which has already been allocated for the insertion of the RH. However, given its large surface, it is unlikely that the entire area correlates solely with the RH insertion. We allocate the posteroventral portion of medial surface of the distal end of the scapular blade in *S. validum* to the SRP; the anterodorsal portion is allocated to the RH (Fig 1).

**Deltoideus scapularis (DSC).** In crocodilians, the deltoideus scapularis originates on a broad area of the lateral surface of the distal half of the scapula. The avian homolog is highly modified and originates more proximally on the acromion process [35, 37], meaning that it can no longer act as a major abductor of the humerus and must be compensated for by a highly developed supracoracoideus [35]. This morphology is not reported in any non-avian dinosaur myological reconstruction [21, 35, 37, 38, 43–45] and so it is much more likely that the DSC retained its ancestral archosaur morphology in *S. validum*. This interpretation is further supported by the expanded distal end of the scapula and its large lateral surface area. There are minor striations visible here in UALVP 2 that likely correspond to the origin of the DSC (Fig 1). It is therefore reconstructed at this location in *S. validum*.

In the EPB, the DSC inserts on the lateral surface of the humerus proximal to the insertion of the deltoideus clavicularis [31, 44]. In birds, this attachment is found on the lateral surface of the deltopectoral crest; it attaches more proximal in crocodilians [23, 35]. The deltopectoral crest bears striations on its lateral surface in ornithischians such as *Heterodontosaurus tucki* Crompton & Charig 1962, *Scutellosaurus lawleri* Colbert 1981, *Kentrosaurus aethiopicus* Hennig 1915, *Stegosaurus stenops* Marsh 1877, *Panoplosaurus mirus* Lambe 1919, *Euoplocephalus tutus* Lambe 1910, *Chasmosaurus belli* Lambe 1914, *Centrosaurus apertus* Lambe 1904, and *Styracosaurus albertensis* Lambe 1913, and is therefore acknowledged as the insertion area for the deltoid muscles [44, 45]. Sauropods, including *D. schrani* and *O. skarzynskii*, likewise display scarring on this part of the humerus which is also attributed to the DSC and deltoideus clavicularis in muscular reconstructions [21, 46]. However, other reconstructions, such as that of *Diamantinasaurus matildae*, place this muscle in a more proximal position on the proximolateral humeral head, similar to that seen in crocodilians [43]. This arrangement is similar to reconstructions of the theropod genus *T. hallae* in which the DSC inserts in a small oval depression just distal to the greater tubercle [35]. UALVP 2 displays no such feature but does bear a distinct pitted texture on the lateral surface of the deltopectoral crest. *Stegoceras validum* is therefore reconstructed here with the basal ornithischian condition in which the DSC inserts on the proximal half of this area (Fig 2). Assuming this morphology, the DSC would have abducted and retracted the humerus [35].

**Deltoideus clavicularis (DCL).** The deltoideus clavicularis has been highly modified in birds to form the propatagialis, which functions in elbow flexion of the wing [35, 36, 47]. Some birds and all crocodilians share an origin for these homologous muscles on the acromial ridge of the scapula [35]. Previous myological reconstructions of a variety of dinosaurs also place the origin for the DCL along this ridge [21, 35, 43, 45]. The acromial ridge of UALVP 2 is well-defined and highly pitted, suggesting that this was indeed the site of origin for the DCL in *S. validum*. Jasinoski *et al.* (2006) extend the origin of this muscle to the acromial depression [37]. However, UALVP 2 possesses only a minor depression with no observable muscle

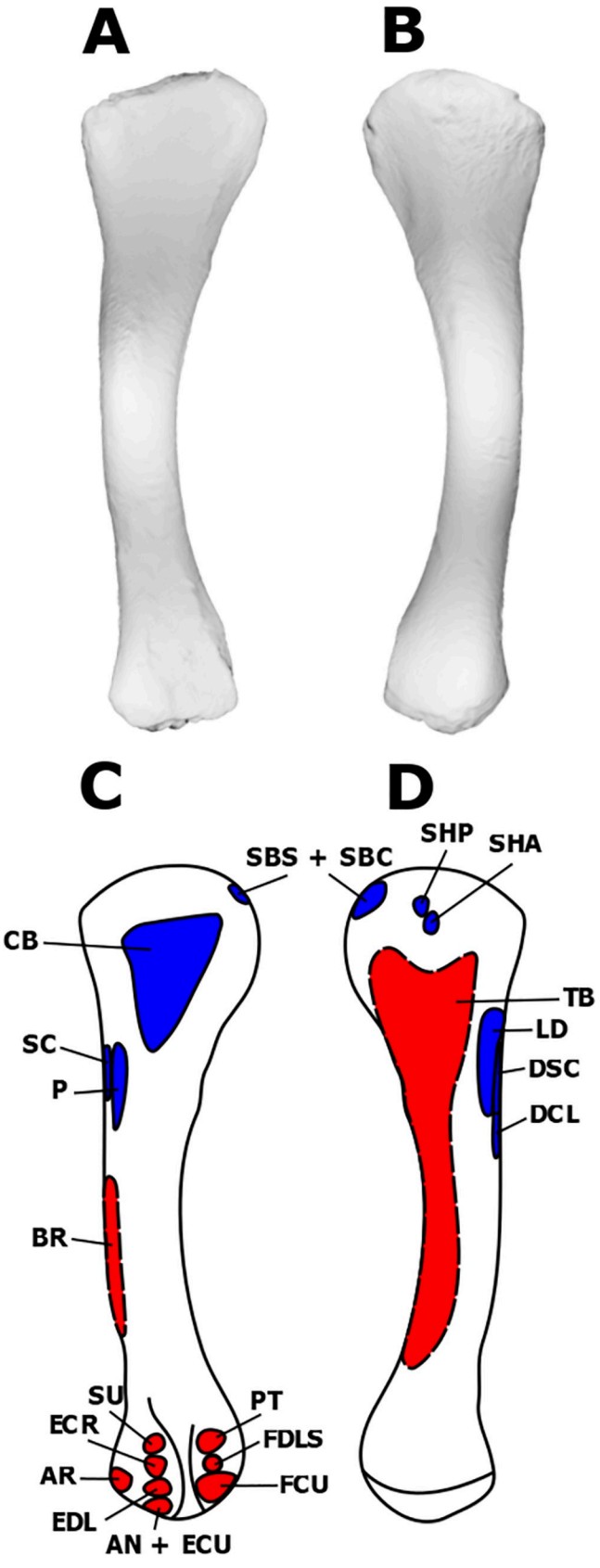

**Fig 2. Myological reconstructions and scans of the right humerus of *Stegoceras validum*.** (A) Anterior scan, (B) posterior scan, (C) anterior reconstruction, and (D) posterior reconstruction. Areas of red indicate muscle origins. Areas of blue indicate muscle insertions. Dashed lines indicate muscles that were reconstructed entirely based on EPB comparisons.

scarring, so it is unlikely that the origin of this muscle extended beyond the acromial ridge in *S. validum* (Fig 1).

As already discussed, the DCL and its homolog in birds inserts on the lateral surface of the deltopectoral crest just distal to the insertion of the DSC in the EPB [21, 33, 35, 44]. This area displays osteological correlates across a variety of dinosaurs including basal ornithischians, stegosaurs, ceratopsids, ankylosaurs, theropods, sauropods [21, 35, 37, 45], and UALVP 2. As such, the DCL is reconstructed on the lateral surface of the deltopectoral crest for *S. validum* (Fig 2). With such a morphology, the DCL would have abducted the humerus in *S. validum* [35].

**Subscapularis (SBS).** The subscapularis originates from the medial surface of the scapular blade in extant archosaurs [35]. Exactly where it originates on the medial surface in extant archosaurs is debated. Jasinoski *et al.* (2006) argue for an origin along the proximal half of the scapular blade on the basis of a ridge in this area that defines the dorsal edge of the SBS. However, this ridge is also present in crocodilians and instead defines the origin of the scapulohumeralis posterior [31]. Burch (2014) argues that the ventrally shifted ridge in *T. hallae* would cause too much of a reduction in surface area for the attachment of the SBS and instead proposes an origin along the flaring blade of the scapula as in crocodilians. In sauropods, it has been proposed that the SBS and subcoracoideus are divisions of the subcoracoscapularis which originates from a raised knob on the medial side of the proximal half of the scapula [21, 34, 46]. UALVP 2 displays no ridges or knobs on the medial surface of the scapula; however, it does display a flared scapular blade. As such, the SBS is reconstructed for *S. validum* here directly proximal to the RH and SRP along the medial surface of the scapular blade (Fig 1).

The insertion of the SBS in the EPB is tendinous and shared with the subcoracoideus. The tendon inserts unequivocally on the medial tuberosity of the humerus [31, 35, 37, 44]. This area often displays well-defined osteological correlates, ranging from a prominent tuberosity to pitted or striated textures, across a wide array of dinosaurs and is attributed to this same insertion [21, 35, 37, 38, 43, 45]. The humerus of UALVP 2 does not display any obvious tuberosity but does have a highly rugose medial corner of the proximal surface of the humerus. This area is therefore attributed to the tendinous insertion of the SBS and subcoracoideus in *S. validum* (Fig 2). This muscle would have retracted and rotated the humerus [35].

**Subcoracoideus (SBC).** The subcoracoideus is fused to the SBS in crocodilians and has no independent osteological attachments [31, 35]. Birds and lepidosaurs retain independent SBCs; however, there is some variation in its structure. In the majority of neognath birds, this muscle is divided into two heads, the caput dorsale and caput ventrale [37]. Some species, however, lack the ventral head [29, 37, 48]. The dorsal head originates from the medial side of the coracoid, overlapping the coracoid foramen [37] while the ventral head originates from the anterior edge of the coracoid [37]. There are no osteological correlates around these areas in UALVP 2. Due to the variability of this muscle in the EPB and the lack of osteological correlates in UALVP 2, it is tentatively reconstructed here having only the dorsal head and an origin surrounding the coracoid foramen (Fig 3). As discussed previously, the insertion for the SBC would have been tendinous on the medial corner of the proximal surface of the humerus. These attachment sites would have resulted in a muscle that adducted and laterally rotated the humerus [35].

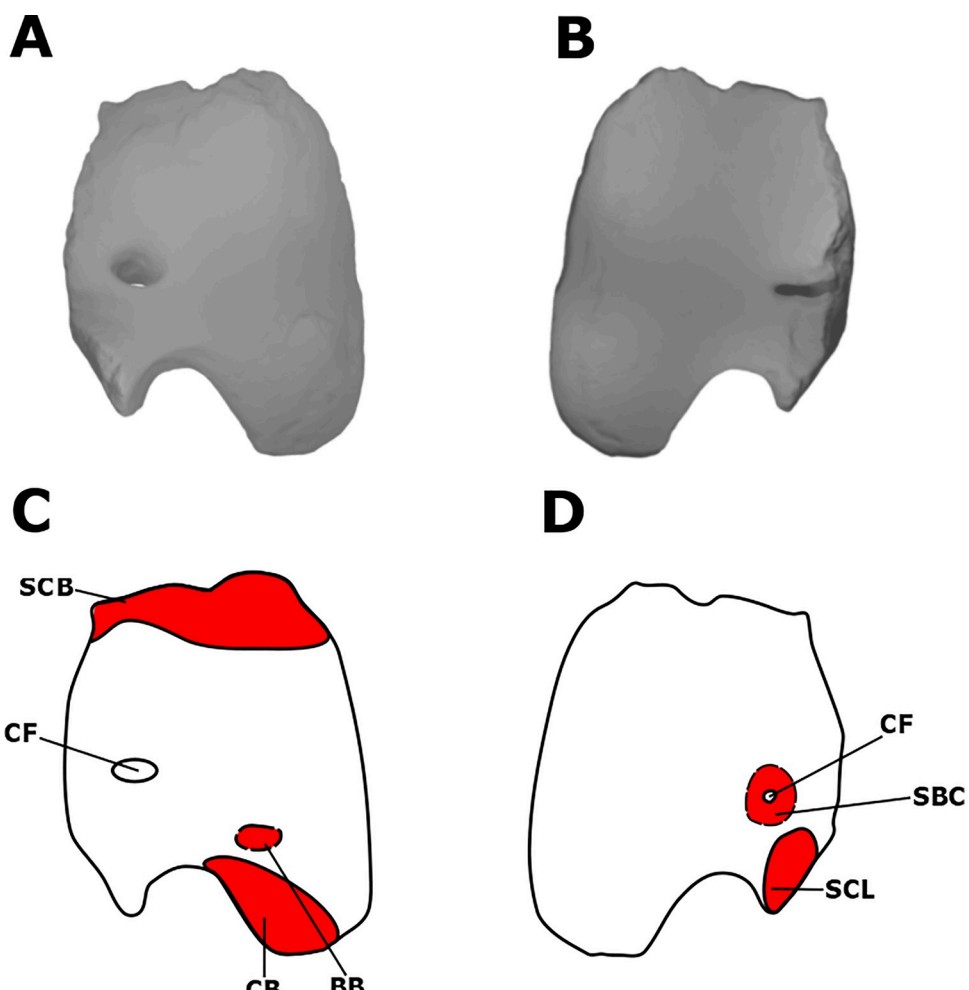

**Fig 3. Myological reconstructions and scans of the right coracoid of *Stegoceras validum*.** (A) lateral scan, (B) medial scan, (C) lateral reconstruction, and (D) medial reconstruction. Areas of red indicate muscle origins. Dashed lines indicate muscles that were reconstructed entirely based on EPB comparisons.

**Scapulohumeralis posterior (SHP).** The scapulohumeralis posterior is found in both crocodilians and birds [21, 35, 37]. In both EPB clades, the muscle originates on the lateral side of the posteroventral edge of the proximal half of the scapular blade. In birds however, the muscle usually extends far more distally, although in *Struthio*, the origin is much more like that of crocodilians [35, 37]. The crocodilian SHP also extends to the medial side of the scapula, inserting ventral to its medial ridge [31]. This same ridge is present in many theropods, and so the origin of the SHP likely extended to the medial surface in these taxa [35]. No such ridge is present in UALVP 2. As such, the SHP is conservatively shown here as originating only from the lateral surface of the scapular blade where the origins of the EPB taxa overlap (Fig 1). This is supported by a mild rugose texture in this area on the scapula of UALVP 2.

The insertion of the SHP is found on the posterior surface of the humerus just distal to the internal tuberosity in the EPB [31, 37]. It can be extensive in crocodilians but is reserved in birds [31, 35, 37]. It has been reconstructed in this area with the conservative morphology of birds in theropods and basal ornithischians, based on striated or rugose texturing [35, 37, 44]. The area just distal to the internal tuberosity of UALVP 2 is heavily pitted, supporting the

notion that it would have served as the insertion site for the SHP in *S. validum* (Fig 2). With this conservative morphology, the scapulohumeralis posterior would have retracted the humerus [44].

**Scapulohumeralis anterior (SHA).** The scapulohumeralis anterior is present in birds but not in crocodilians [21, 35, 37]. It has been hypothesized that modern crocodilians may have apomorphically lost the muscle and may not exhibit the ancestral archosauromorph condition [21, 49–51]. As such, myological reconstructions of this muscle in dinosaurs have largely been based on its presence in birds and lepidosaurs [21, 34, 35, 37]. In these taxa, the SHA originates, at least partially, on the posteroventral margin of the scapular blade at its most proximal extent [21, 34, 35, 37]. In lepidosaurs, this origin also extends to the lateral surface of the scapula [21, 23]. On the scapula of UALVP 2, there is a slight rugosity corresponding to the area of origin for the SHA in birds. This rugosity, however, cannot be differentiated from that attributed to the SHP. It is likely that the SHA originated in *S. validum* from the proximal-most extent of the posteroventral margin of the scapular blade, just proximal to the SHP, and that the observed rugosity corresponds to both muscles (Fig 1).

In birds, the tendinous insertion for the SHA is found just distolateral to that of the SHP. It is denoted in the EPB by the pneumatic fossa, which is not present in dinosaurs [35, 37]. Theropod reconstructions of this muscle are based entirely on the EPB and do not display any osteological correlates [35, 37]. Some sauropods display a bulge on the proximolateral edge of the deltopectoral crest that has been attributed to the insertion of the SHA [21, 38] while others display a rugose texture in this same area [52]. UALVP 2 displays no such features but does have a rugose texture directly distolateral to the insertion of the SHP. As such, we propose that the SHA attached to the posterior surface of the humerus, distolateral to the internal tuberosity, as is the condition in birds and non-avian theropods (Fig 2). Similar to its posterior counterpart, the SHA would have retracted the humerus [35].

**Supracoracoideus (SC).** The supracoracoideus originates primarily from the coracoid in the EPB but attaches to the scapula in crocodilians and to the sternum in neognath birds [35]. Additionally, crocodilians have three well-developed heads (supracoracoideus longus, intermedius, and brevis) of the SC while birds have only one [31]. The supracoracoideus longus (SCL) originates on the medial surfaces of the scapula and coracoid along their posteroventral and articular margins [31]. On the lateral surface, the supracoracoideus brevis (SCB) arises from the coracoid dorsal to the coracoid foramen, and the supracoracoideus intermedius (SCI) arises along the dorsal edge of the suture between the scapula and coracoid [31]. In theropods, it has been proposed that this muscle complex minimally originates from the coracoid and mainly attaches to the subacromial depression of the scapula [35, 37]. Voegele *et al.* (2020) propose a more conservative scapular attachment in sauropods confined to the anterior margin of the scapula, based on striations on the coracoid next to the scapulocoracoid suture. This attachment is consistent with the myological reconstruction of basal ornithischians by Maidment & Barrett (2011). Maidment & Barrett (2012) propose a more extensive attachment for the SC in stegosaurs on the large anterolateral surface of the acromial process and dorsal coracoid. Likewise, ceratopsids are also proposed to have a large attachment for the SC on the anterolateral scapula and dorsal coracoid [45]. Ornithopods including *Hypsilophodon foxii* Huxley 1870, and *Lambeosaurus lambei* Parks, 1923 have been reconstructed with the SC mainly attaching to the dorsolateral surface of the anterior scapula, and minimally attaching to the coracoid, if at all [45]. Unfortunately, the coracoid suture of the scapula is not fully preserved in UALVP 2 and what is preserved does not display any obvious osteological correlates. The subacromial depression, however, is fully preserved and displays no evidence for the attachment of the SC. Though lack of an osteological correlate does not rule out attachment of the

SC at the subacromial depression, we reconstruct this muscle as attaching to the scapula along the scapulocoracoid sutureas it does in crocodilians (Fig 1) [21, 31, 35, 37, 44].

The coracoid origin of the SC is difficult to locate in *S. validum*. All the margins of the element, save for part of the ventral margin, are mostly broken. Where preserved, striations are visible along the lateral face of the dorsal margin, possibly corresponding to the origin of the SCB. Minimal striations are also preserved on the medial surface of the ventral margin of the element, bordering on the scapulocoracoid suture. The area corresponds with part of the origin of the SCL in crocodilians [31]. As such, these areas have been reconstructed as origin sites for the SC complex in *S. validum* (Fig 3). There are no visible osteological correlates for the SCI on UALVP 2, as the part of the bone that would potentially support this muscle is not preserved. Therefore, the SCI is excluded from the myological reconstruction here.

The insertion of the SC is equivocal in the EPB. Crocodilians display a tendinous insertion on the apex of the deltopectoral crest whereas birds have a tendinous insertion on the greater tubercle [31, 35, 37, 44]. The avian attachment has been highly modified, as the SC functions in the elevation and rotation of the wing during the upstroke [35, 53]. As such, it is highly unlikely that the SC would display such a modification in non-avian dinosaurs. Burch (2014) instead reconstructed the SC of *T. hallae* as inserting along the proximolateral edge of the deltopectoral crest on the basis of a small ovoid depression. The insertion of the SC has also been reconstructed in the area for a wide variety of ornithischian taxa [45]. The humerus of UALVP 2 possesses the same feature described by Burch (2014) with a highly rugose surface texture. The SC of *S. validum* therefore likely inserted at this same area (Fig 2). Such a morphology would result in a muscle that protracted the humerus [35].

**Coracobrachialis (CB).** The coracobrachialis is well-known in the EPB and unequivocal in dinosaurs [21, 23, 31, 34, 35, 37, 44, 45]. Its origin is located on the posteroventral portion of the lateral surface of the coracoid in both crocodilians and paleognath birds [35, 37]. It has been reconstructed in this general position in theropods [35, 37], sauropods [21, 38, 43, 54], and ornithischians [44, 45]. UALVP 2 displays notable striations in this same area which are attributed to the CB origin (Fig 3).

The insertion of the CB is also consistent in the EPB and is found on the anterior face of the humerus distal to the proximal articular surface [35]. It extends distally, reaching the medial surface of the deltopectoral crest [35]. In theropods, this area is defined by a large triangular depression with a distally facing apex, which is attributed to the insertion of the CB [35]. This same morphology and proposed insertion are present in sauropods and ornithischians [21, 44, 45]. UALVP 2 displays a similar depression with rugose scarring on its proximal half and faint striations on its distal half. The medial surface of the deltopectoral crest also displays a heavily pitted texture. Based on these osteological correlates and the EPB condition [21, 35, 44, 45], this area is attributed to the insertion of the CB in *S. validum* (Fig 2). This muscle would have protracted the humerus.

**Triceps brachii (TB).** The triceps brachii is present in all archosaurs but the number of heads varies in the EPB [21, 23, 31, 34, 35, 37, 44, 46]. Both birds and crocodilians have the scapular and medial heads, but birds have a vestigial coracoid head and have lost the lateral head [35]. Across all archosaurs and lepidosaurs, the scapular head has a small tendinous origin posterodorsal to the scapular contribution to the glenoid fossa, often associated with a rugose tubercle [21, 35, 37, 38, 55]. This area of the scapula is partially broken on UALVP 2. However, the uppermost margin of a rugose tubercle is preserved, suggesting that *S. validum* displayed a similar scapular origin of the TB (Fig 1). Unfortunately, the extent of the attachment area is impossible to know.

The medial head of the TB originates on the posteromedial surface of the humeral shaft in both birds and crocodilians [31, 35, 37]. It is a fleshy attachment that bifurcates proximally,

around the SHP in crocodilians and SHA in birds, and almost completely covers the humeral shaft [35, 37, 44]. This area lacks osteological correlates in dinosaurs [21, 35, 37, 44, 45], including UALVP 2. As such, the medial head of the TB is reconstructed here as covering the posteromedial humeral shaft based entirely on EPB comparisons (Fig 2). UALVP 2 displays no osteological correlate for the lateral head of the TB. Because there is no direct evidence for the presence of the TB lateralis in UALVP 2, and this muscle is equivocal in the EPB, it is not reconstructed here in *S. validum*.

The heads of the TB share a single tendinous insertion on the posterior surface of the olecranon process of the ulna in the EPB [21, 35, 37, 44, 45]. Most dinosaurs display some form of osteological correlate on the olecranon process that has been attributed to the insertion of the TB [21, 35, 36, 44]. UALVP 2 also displays distinct striations on the olecranon process that are attributed here to the tendon of the TB (Fig 4). The muscle would have functioned in extending the humerus and antebrachium [35].

**Biceps brachii (BB).**   The primary head of the biceps brachii arises from the coracoid in the EPB. Its tendinous origin attaches to a tubercle anterior to the glenoid fossa in both birds and crocodilians [35]. Such a tubercle is also present in theropod dinosaurs and is assumed to be the site of origin [35, 37]. Many sauropods display a ridge at this same area instead of a tubercle, which has also been attributed to the origin of the BB [21, 34, 38, 56–59]. Basal ornithischians (e.g., *Lesothosaurus diagnosticus*) are reported as displaying no such feature by Maidment & Barrett (2011); however, many more derived ornithischian taxa (e.g., hadrosaurs and ankylosaurs) display a tubercle or ridge associated with the BB [45]. In UALVP 2, no such osteological correlates are visible on the lateral surface of the coracoid. Given that the coracoid origin of this muscle is almost unanimously reconstructed in the same position as the EPB in dinosaurs, it is reconstructed here for *S. validum* in the same position, anteroventral to the coracoid foramen and dorsal to the origin of the CB (Fig 3). Birds and non-avian theropods also possess a humeral head of the BB, which is denoted by a rounded area on the anterior surface of the internal tuberosity [35, 37]. This feature is not present in ornithischians [44, 45] nor UALVP 2, and so is not reconstructed for *S. validum*.

The tendinous insertion of the BB is equivocal in the EPB [21, 23, 31, 34, 35, 37, 44]. Birds display two insertions for this muscle, one on the proximal end of the radius and the other on the ulna. Crocodilians, however, possess only the radial insertion [21, 31, 35, 37]. The ulnar insertion of the BB has been reconstructed in theropods and ornithischians, based on EPB comparisons and varying osteological correlates [35, 44, 45]. Its presence varies in sauropods [21, 34, 38, 43, 54]. The radius of UALVP 2 shows distinct striations on the proximal portion of its posterolateral surface that likely corresponded to this muscle. The ulna, however, shows no distinct osteological correlate for the insertion of the BB. It is therefore conservatively reconstructed here as only inserting on the posterolateral surface of the proximal radius of *Stegoceras validum* (Fig 5); however, the lack of an ulnar osteological correlate does not rule out an ulnar insertion. The muscle would have flexed the antebrachium [35].

**Pectoralis (P).**   The origin of the pectoralis involves a variety of elements in archosaurs [35]. In the EPB, one common origin is on the ventral face of the sternum. Additional origins are found on the sternal ribs in crocodilians and the coracoid in birds [35, 37]. There is no evidence for an origin on the coracoid of UALVP 2. Without elements of the sternum preserved in UALVP 2, it is also impossible to determine if this was indeed the origin site for the P in *S. validum*.

The insertion of the P is found on the apex of the deltopectoral crest of both birds and crocodilians [21, 31, 35, 37, 44]. Significant rugose scarring is visible on the deltopectoral apex in UALVP 2, making it likely that this was also the case in *S. validum*. The P is therefore

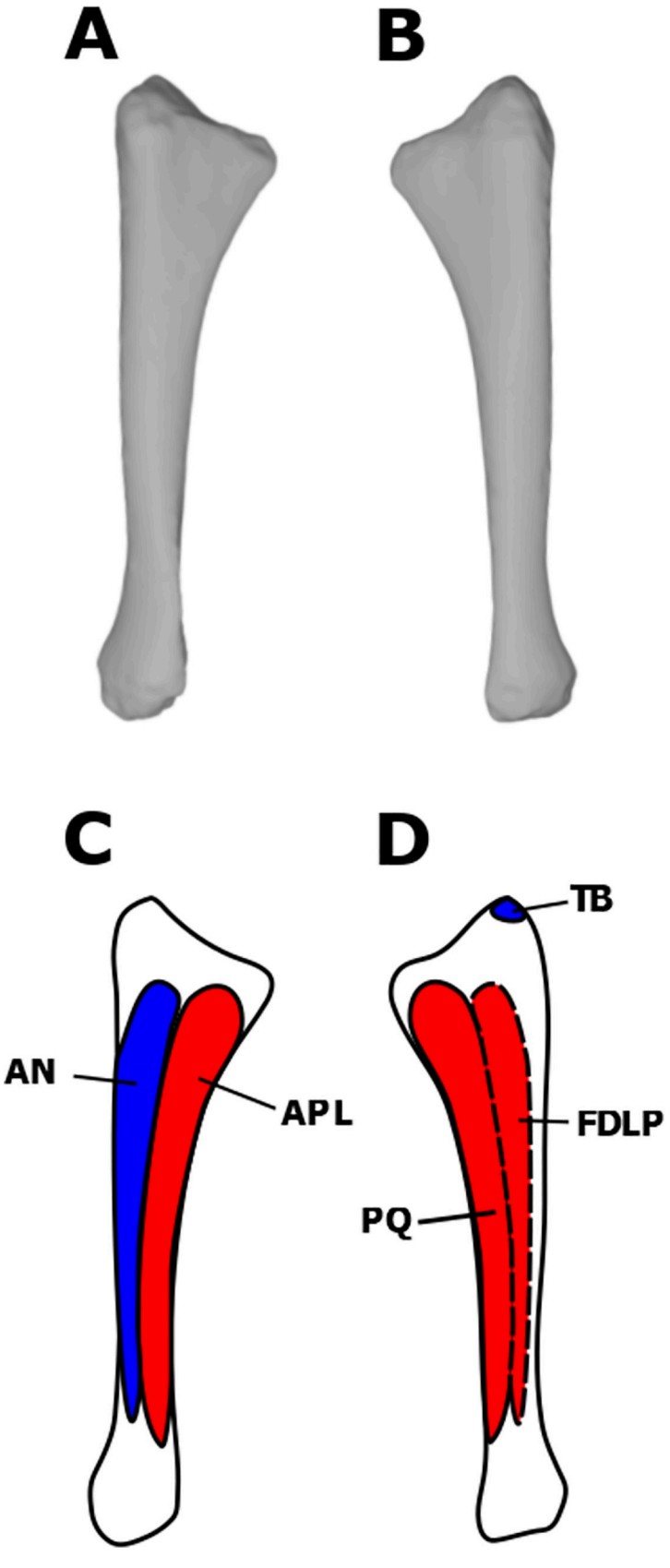

**Fig 4. Myological reconstructions and scans of the right ulna of *Stegoceras validum*.** (A) Lateral scan, (B) medial scan, (C) lateral reconstruction, and (D) medial reconstruction. Areas of red indicate muscle origins. Areas of blue indicate muscle insertions. Dashed lines indicate muscles that were reconstructed entirely based on EPB comparisons.

reconstructed here as inserting on the apex of the deltopectoral crest (Fig 2). Assuming an origin from a sternum or sternal ribs, this muscle would have adducted the humerus in *S. validum*.

**Humeroradialis (HR).**　The humeroradialis of crocodilians is homologized with the propatagialis of birds, based on them sharing an embryological origin [31, 35]. The HR of crocodilians originates from an area distolateral to the deltopectoral crest [31, 35, 37]. Osteological correlates have been found near this area on the humeri of theropods and sauropods, resulting in reconstructions displaying the crocodilian morphology [21, 35, 37]. Maidment & Barrett (2011) elected to exclude this muscle from their reconstruction of basal ornithischian locomotor musculature due to its questionable homology and a lack of osteological correlates. UALVP 2 also displays no osteological correlate of the HR on the humerus.

The insertion of the HR is marked by a distinct tubercle on the posterior surface of the radius in crocodilians [31]. No such osteological correlate is visible in UALVP 2. Due to a lack of evidence and an unresolved homology for the archosaurian HR, it is not included here in the myology of *S. validum*.

**Brachialis (BR).**　The brachialis in crocodilians arises from an elongate region on the anterolateral surface of the humeral shaft, distal to the deltopectoral crest and proximal to the distal condyles [31, 35, 44]. This condition is also seen in lepidosaurs and turtles [60, 61]. Birds possess the fossa musculus BR on the anterior surface of the humerus, just proximal to the condyles, for the attachment of this muscle [28, 35, 44]. Although the humerus of UALVP 2 displays a broad anterior intercondylar depression, there is no evidence for a distinct fossa as seen in birds. There are also no osteological correlates along the anterolateral margin of the humerus, distal to the deltopectoral crest. Because most dinosaurs are conservatively reconstructed with the assumed primitive crocodilian morphology [21, 35, 37, 44, 45], this structure is also assumed here for *S. validum* (Fig 2).

The BR inserts with the BB on the proximal ends of the radius and ulna in birds, but only on the radius in crocodilians [21, 31, 35, 44]. As with the BB, there is no evidence for an ulnar insertion in *S. validum*. The BR is therefore reconstructed as inserting with the BB on the proximal end of the radius in *S. validum* (Fig 5). With this morphology, the muscle would have flexed the forearm [35].

**Latissimus dorsi (LD).**　The latissimus dorsi arises from the neural spines of the last cervical vertebra and the first six or seven dorsal vertebrae in the EPB [31, 35, 44]. It is thus inferred to have the same origin in *S. validum*.

The insertion of the latissimus dorsi is found on the proximoposterior surface of the humeral shaft, lateral to the midline in both birds and crocodilians [31, 44, 47]. An ovoid rugose scar is observed in this area on the humerus of UALVP 2. Similar scars have been observed in *Ankylosaurus magniventris* Brown, 1908 [45, 62], dromaeosaurids [37], *Maiasaura peeblesorum* [29], *Mantellisaurus atherfieldensis* [63], sauropods [34, 38, 55, 58], and stegosaurs [44, 45]. Based on the observed osteological correlates in a variety of dinosaurs, the insertion in the EPB taxa, and the scar present in UALVP 2, the insertion of the latissimus dorsi is located on the proximolateral portion of the posterior surface of humeral shaft in *S. validum* (Fig 2). Such a morphology means that this muscle would have retracted the humerus [35].

**Teres major (TM).**　The teres major occurs in crocodilians but not in birds [36, 37, 64]. In crocodilians, its origin is located on the lateral surface of the posterodorsal scapular blade

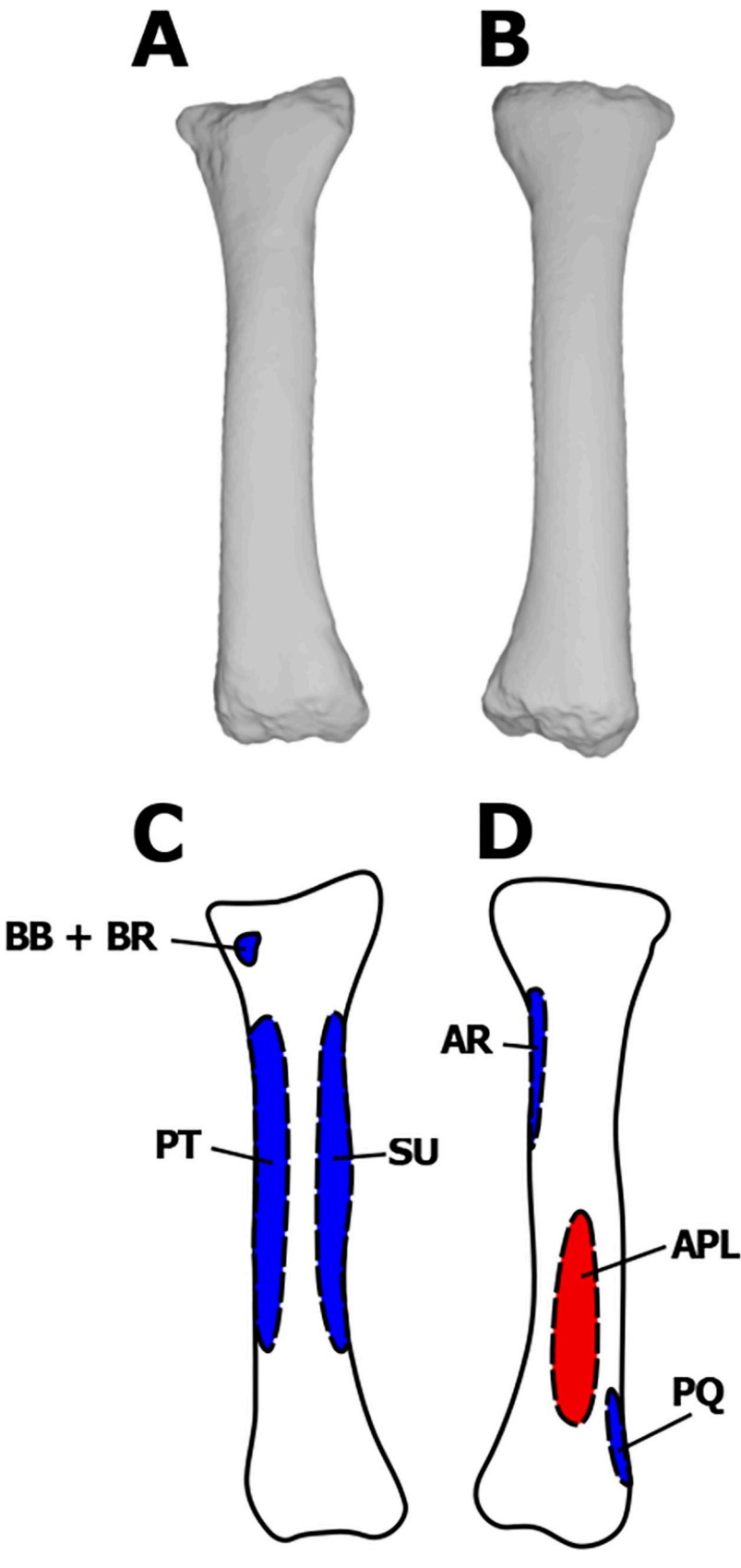

**Fig 5. Myological reconstructions and scans of the right radius of *Stegoceras validum*.** (A) Lateral scan, (B) medial scan, (C) lateral reconstruction, and (D) medial reconstruction. Areas of red indicate muscle origins. Areas of blue indicate muscle insertions. Dashed lines indicate muscles that were reconstructed entirely based on EPB comparisons.

ventral to the DSC, and its insertion is shared with the latissimus dorsi on the proximal humerus. There is no visible scarring corresponding the origin of the TM on the scapular blade of UALVP 2. Because there is no observable osteological correlate for this muscle in UALVP 2 and it is not present in all EPB taxa, it is not reconstructed here in *S. validum*.

## Antebrachial musculature

**Anconeus (AN).** The anconeus muscle originates on the ectepicondyle of the humerus in the EPB [35]. It is referred to as the ectepicondylo-ulnaris in birds [65] and as the flexor ulnaris in crocodilians by Meers (2003). Of the muscles that attach to the ectepicondyle, the AN attaches the most distally [35]. Ancestral Aves have a fused tendinous origin for the AN and extensor carpi ulnaris [35]. Crocodilians lack the extensor carpi ulnaris entirely [31, 35]. Squamates also lack the extensor carpi ulnaris and turtles display a similar fusion to that of birds [35, 62, 66]. This inconsistency in modern reptiles makes it difficult to determine which morphology was more likely in non-avian dinosaurs. The ectepicondyle of UALVP 2 is highly pitted with no way of differentiating individual muscle scars. As such, both the AN and extensor carpi ulnaris are reconstructed here for *S. validum* as attaching to the distal portion of the humeral ectepicondyle (Fig 2). However, differentiating the origins of these muscles is impossible.

The insertion of the AN is found on the lateral surface of the ulna in archosaurs [35]. The insertion extends the entire length of the ulnar shaft in the EPB [35]. Prominent striations on the lateral surface of the ulna at its proximal end support such an attachment in *S. validum* (Fig 4). This muscle would have flexed the forearm [35].

**Extensor carpi ulnaris (ECU).** As discussed previously, the extensor carpi ulnaris is tentatively reconstructed here as attaching to the distal ectepicondyle of the humerus, along with the AN (Fig 2). The ECU has multiple tendinous insertions on various carpals and metacarpals in the EPB [35]. These attachments vary in modern reptiles and there is no way to tell how this muscle may have inserted in *S. validum* without preserved carpals and metacarpals. As such, the insertion points of the ECR remain uncertain in *S. validum* and are not reconstructed here.

**Supinator (SU).** The supinator has the most proximal origin of the muscles that attach to the ectepicondyle in crocodilians [35, 66]. In birds, it attaches more distally, and the proximal position is taken by the extensor carpi radialis [35]. The avian positioning is specialized to aid in flight adjustments of the wing [35, 67], so it is likely that the ancestral archosaurian condition is more like that of crocodilians. Once again, it is impossible to separate the muscle attachments to the ectepicondyle in UALVP 2, so the origin the SU in *S. validum* is assumed to be the same as that observed in crocodilians (Fig 2).

The insertion of the SU is located on the anterolateral surface of the radius in the EPB, adjacent to that of the pronator teres [29, 31, 35]. It usually covers almost the entire length of the radial shaft [35]. Although there are no correlates for this muscle on the radius of UALVP 2, it is reconstructed here with the same morphology as the EPB in *S. validum* (Fig 5). This muscle would have flexed the forearm [35].

**Extensor carpi radialis (ECR).** In crocodilians, the extensor carpi radialis arises from the ectepicondyle of the humerus between the SU and extensor digitorum longus [35]. Birds display a more proximal origin [28, 35]. This positioning serves to better flex and extend the wrist and elbow of the avian wing during flight [35, 67]. It is unlikely that non-avian dinosaurs had a

use for such an adaptation, so it is assumed that the origin of the ECR of *S. validum* resembled that of crocodilians (Fig 2).

In crocodilians, the ECR inserts on the radiale [31]. In birds, it inserts on the carpometacarpus on metacarpal I [35]. Unfortunately, there is no preserved radiale or metacarpal I in UALVP 2. Because the avian wrist has been highly modified and non-avian dinosaurs display a morphology much closer to that of crocodilians, it is assumed that the ECR of *S. validum* would have inserted on the radiale as it does in crocodilians. This muscle would have flexed the forearm, and extended and adducted the wrist [35].

**Extensor digitorum longus (EDL).** The origin of the EDL shows little variation in the EPB, almost always arising from the middle of the ectepicondyle of the humerus between the ECR and AN [35]. As previously discussed, the entire ectepicondyle of UALVP 2 is highly rugose. As such, the EDL is reconstructed here as originating between the ECR and SU, as it does in extant archosaurs (Fig 2).

The EDL consistently inserts via tendons on the base of metacarpals I-IV in the EPB [35]. Because the metacarpals of UALVP 2 are not preserved, it is impossible to determine if this was the case in *S. validum*. It is assumed here that *S. validum* would have shared the same condition as all extant archosaurs.

**Abductor radialis (AR).** The abductor radialis originates lateral to the ECR on the ectepicondyle of the humerus in crocodilians [31, 35]. This muscle is not present in birds but is sometimes homologized with the second head of the ECR observed in some species [29, 35]. The ectepicondyle of the humerus of UALVP 2 is generally rugose, and so the exact attachments of different muscles to this structure are indistinguishable and must be inferred from EPB comparisons. Due to the modifications for flight in the muscles of birds, we infer that *S. validum* possessed an independent AR that arose from a similar position as that of crocodilians (Fig 2).

The AR has a shared tendinous insertion with the ECR on the carpometacarpus of birds, and an independent insertion on the proximal half of the radial shaft medial to the SU in crocodilians [31, 35]. Given the highly modified avian wrist morphology, it is unlikely that their associated muscular attachments were shared by non-avian dinosaurs. Although there are no osteological correlates for the AR on the radius of UALVP 2, we infer that the musculature of *S. validum* would have been more similar to that of crocodilians. As such, the AR is reconstructed as inserting on the entire lateral surface of the proximal half of the radius in *S. validum* (Fig 5). This morphology would result in a muscle that abducted the forearm [35].

**Abductor pollicis longus (APL).** The abductor pollicis longus (extensor longus alulae in birds) is found in both birds and crocodilians. It has two heads: one originating from the from the lateral ulnar shaft anterior to the AN, and the other from the distomedial radial shaft [29, 31, 35]. The ulnar head usually attaches to the full length of the shaft, and the radial head only attaches to the distal half [35]. There are no osteological correlates present on the radial shaft of UALVP 2, but there are minor striations present on the proximal and distal ends of the ulnar shaft. The APL is reconstructed here as having the same origin in *S. validum* as the EPB because of partial osteological correlates (Figs 4 & 5).

The insertion sites for the APL differ between birds and crocodilians. The APL inserts on the developmental equivalent to metacarpal I in birds and on the radiale in crocodilians [31, 35, 68]. The insertion on metacarpal I is also shared with turtles and lepidosaurs [61, 62], making it likely that this is the ancestral condition. Since there is no preserved radiale or metacarpals in UALVP 2, we hypothesize that the insertion of the APL would have been the same in *S. validum* as it is in birds, turtles, and lepidosaurs. The muscle would have therefore have extended and abducted the wrist, and abducted digit I [35].

**Pronator teres (PT).**    The pronator teres is present in most EPB taxa and has the most proximal origin of muscles attaching to the entepicondyle of the humerus [35, 69–71]. As with the ectepicondyle, the entepicondyle of UALVP 2 is highly rugose with no visible differentiation of scar textures. Therefore, the muscles attaching to this structure are entirely based on EPB comparisons. The PT is reconstructed here attaching to the proximal margin of the entepicondyle in *S. validum* (Fig 2).

The insertion for the PT is found on the anteromedial surface of the radius in the EPB [35]. However, the extent of its insertion varies. In all crocodilians and paleognath birds, the muscle inserts over most of the length of the radial shaft. This is likely the ancestral condition. Neognaths, to the contrary, have a shorter insertion on less than half of the proximal radius [35]. With no osteological correlates present on the radial shaft of UALVP 2, we reconstruct the PT as having a long insertion on the anteromedial face of the radius because it is the ancestral archosaur condition (Fig 5).

**Pronator accessories (PA).**    The pronator accessories is absent in all crocodilians and paleognath birds [35, 72]. It is, however, present in neognaths, squamates, and turtles [35]. Unfortunately, this means that its presence in non-avian dinosaurs is uncertain. With no direct evidence for the origin or insertion of this muscle on the humerus or radius of UALVP 2, it has been excluded from the appendicular muscle reconstruction of *S. validum*.

**Pronator quadratus (PQ).**    The pronator quadratus is present in crocodilians and is assumed to be homologous with the ulnimetacarpalis ventralis of birds [21, 31, 35, 44, 47]. It always originates along the medial surface of the ulna; however, its total area of attachment varies [31, 35, 60, 61]. The attachment covers more than half of the ulnar shaft in crocodilians but is restricted to the distal half in most birds [29, 35]. It has been reconstructed with the crocodilian morphology in theropods [35] and sauropods [21]. The ulna of UALVP 2 exhibits striations at its proximal and distal articulations, which extend down onto the proximal and distal ends of the shaft, although the rest of the shaft displays no osteological correlates. It is possible that these striations are related to a proximal origin of the PQ, as seen in crocodilians. As such, we infer that the origin of the PQ attached over the full length of the medial surface of the ulnar shaft (Fig 4).

In crocodilians, the PQ inserts onto the ulnar-facing side of the radial shaft [31, 35, 60]. In birds, the insertion extends to the carpals [35, 61, 73, 74]. The radius of UALVP 2 displays no osteological correlates associated with the PQ; however, we infer that there would have still been an insertion on its lateral surface, based on the EPB. Unfortunately, without preserved carpals, it is impossible to say whether this insertion extended beyond the radius in *S. validum*. It is therefore conservatively constructed here as only being present on the radius (Fig 5). With such a morphology, this muscle would have pronated the antebrachium [35].

**Epitrochleoanconeus (EA).**    Known as the entepicondylo-ulnaris in birds, this muscle is not present in crocodilians but is present in other reptiles, including turtles and lepidosaurs [35, 46]. Among these reptiles, its morphology is highly varied [35]. With little consistency in extant reptiles, the highly modified wing musculature of birds, and no direct osteological correlate for this muscle on UALVP 2, the EA is excluded from this myological reconstruction of *S. validum*.

**Flexor carpi ulnaris (FCU).**    The flexor carpi ulnaris is found in the consulted EPB taxa and has a tendinous origin on the distal portion of the humeral entepicondyle [35]. It has a small second tendinous attachment in many birds, which is usually associated with controlling secondary flight feathers [29, 35]. Because non-avian dinosaurs did not possess flight feathers [3, 9], such a configuration is unlikely to have occurred. The FCU is therefore presented here for *S. validum* with the same morphology observed in crocodilians (Fig 2).

The FCU has a consistent insertion on the pisiform in the EPB [35]. Birds, however, possess an additional insertion on the ulnare, which is not homologous to the ulnare of crocodilians [35, 68]. Because this element is not present in non-avian dinosaurs, the EPB suggests that the FCU of *S. validum* would have solely attached to the pisiform. This morphology would have resulted in a muscle that flexed and adducted the wrist [31].

**Flexor digitorum longus superficialis (FDLS).** The flexor digitorum longus superficialis is present in all crocodilians and birds except ratites [35, 75]. Its origin is tendinous and located between those of the FCU and the PT [35]. With no variability in the EPB taxa, this origin is reconstructed with the same position in *S. validum*, and is associated with the rugosity of the entepicondyle (Fig 2).

The FDLS shares tendinous insertions with the flexor digitorum longus profundus in the EPB [35]. In modern reptiles, including turtles and lepidosaurs, these tendons attach to the ventral surfaces of all the distal phalanges [35, 60, 61]. Crocodilians have reduced manus digits IV and V, and so the FDLS only attaches to the other three phalanges in these animals [31]. Birds have a highly fused and modified manus in which there is only one tendinous attachment for the FDLS on digit II [35, 76]. Given the modified nature of the digits of the EPB, it is difficult to reconstruct this attachment in *S. validum* with no preserved elements of the manus. It is tentatively reconstructed here as having tendinous insertions on the ventral side of all the distal phalanges. The FDLS would have acted to flex the wrist and digits [35].

**Flexor digitorum longus profundus (FDLP).** The presence of the FDLP is consistent in the EPB [35]. It originates over the entire length of the medial ulnar shaft in crocodilians and is posterior to the PQ. In birds, the origin of the FDLP is halted distally by the PQ [29, 35]. This reduction is likely linked to the reduction of the digits in birds, and so less muscle mass was required to flex the digits [35]. There are no osteological correlates on the ulnar shaft that can be specifically attributed to the FDLP in UALVP 2. Therefore, the origin of the FDLP is reconstructed in *S. validum* as having the crocodilian morphology because the manus of pachycephalosaurs was not as highly modified as those of modern birds (Fig 4). As stated previously, the FDLP shares insertions with the FDLS, and is reconstructed here in *S. validum*.

By combining the previous individual muscle reconstructions, we created a 3D model of the pectoral and forelimb musculature of *S. validum* (Fig 6). The model shows how each muscle attaches to the elements of the postcranial skeleton and how the muscles are oriented and overlap with each other in three dimensions.

## Pelvic musculature

**Iliocaudalis (ILC).** The iliocaudalis is found in both birds and crocodilians, and is therefore unequivocally present in dinosaurs [22, 23, 77]. This muscle originates from the lateral surface of the posterior region of the postacetabular process of the ilium in the EPB [22, 23, 77], and has been reconstructed this way for dinosaurs such as *D. schrani* [22], *O. skarzynskii* [38], *Rapetosaurus krausei* Curry Rogers & Forster, 2001 [56], *Hypsilophodon foxii* [45], *Jeholosaurus shangyuanensis* Xu *et al.*, 2000 [45], many hadrosaurs [23, 45], and *Tyrannosaurus rex* Osborn, 1905 [77]. Large quadrupedal ornithischians (e.g., *Chasmosaurus belli* and *Kentrosaurus aethiopicus*) have a reduced iliac attachment area for the ILC due to the presence of a supratrochanteric flange [45]. UALVP 2 displays distinct striations on the posterior postacetabular process of the ilium and does not possess a supratrochanteric flange. As such, we reconstruct the ILC as attaching to the posterior acetabular process of the ilium in *S. validum* (Fig 7).

The ILC shares insertions with the ischiocaudalis on the anterior caudal vertebrae in crocodilians [77]. Attachments are found on the lateral and ventral surfaces of the caudal ribs and

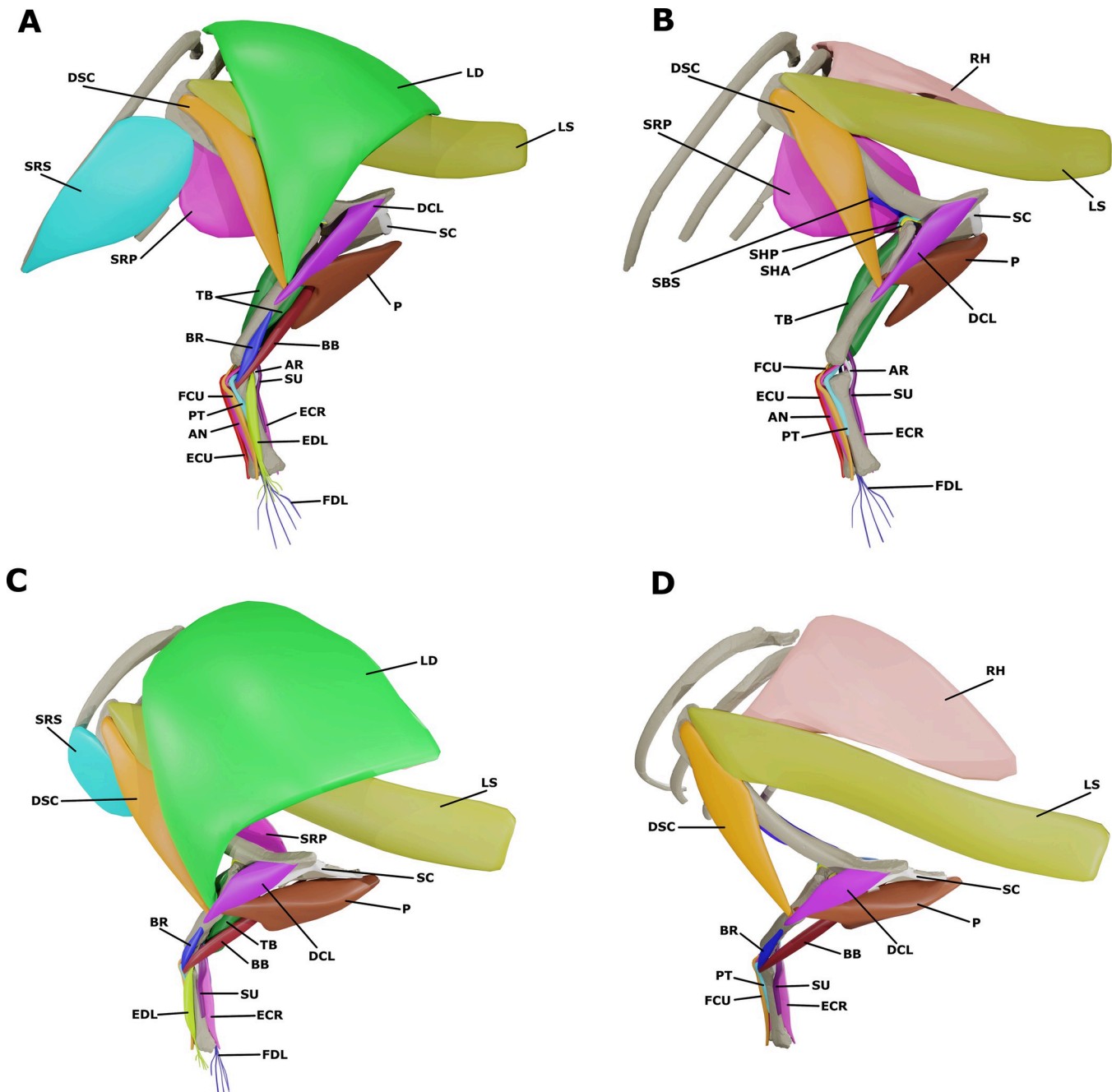

**Fig 6. Pectoral and brachial muscular reconstruction of *Stegoceras validum*.** (A) Superficial musculature in lateral view. (B) Deep musculature in lateral view. (C) Superficial musculature in anterodorsolateral view. (D) Deep musculature in anterodorsolateral view.

on the ventral tips of the haemal spines [77]. Although many caudal vertebrae are preserved in UALVP 2, they either do not display any obvious osteological correlates for these insertions or do not preserve transverse processes or haemal spines. As such, based on the EPB, these attachments were probably present in *S. validum*. 6985

**Ischiocaudalis (ISC).** If the homology of the pubocaudalis of birds to the ischiocaudalis of crocodilians is accepted, then the presence of the ISC in dinosaurs is unequivocal [22, 23, 77].

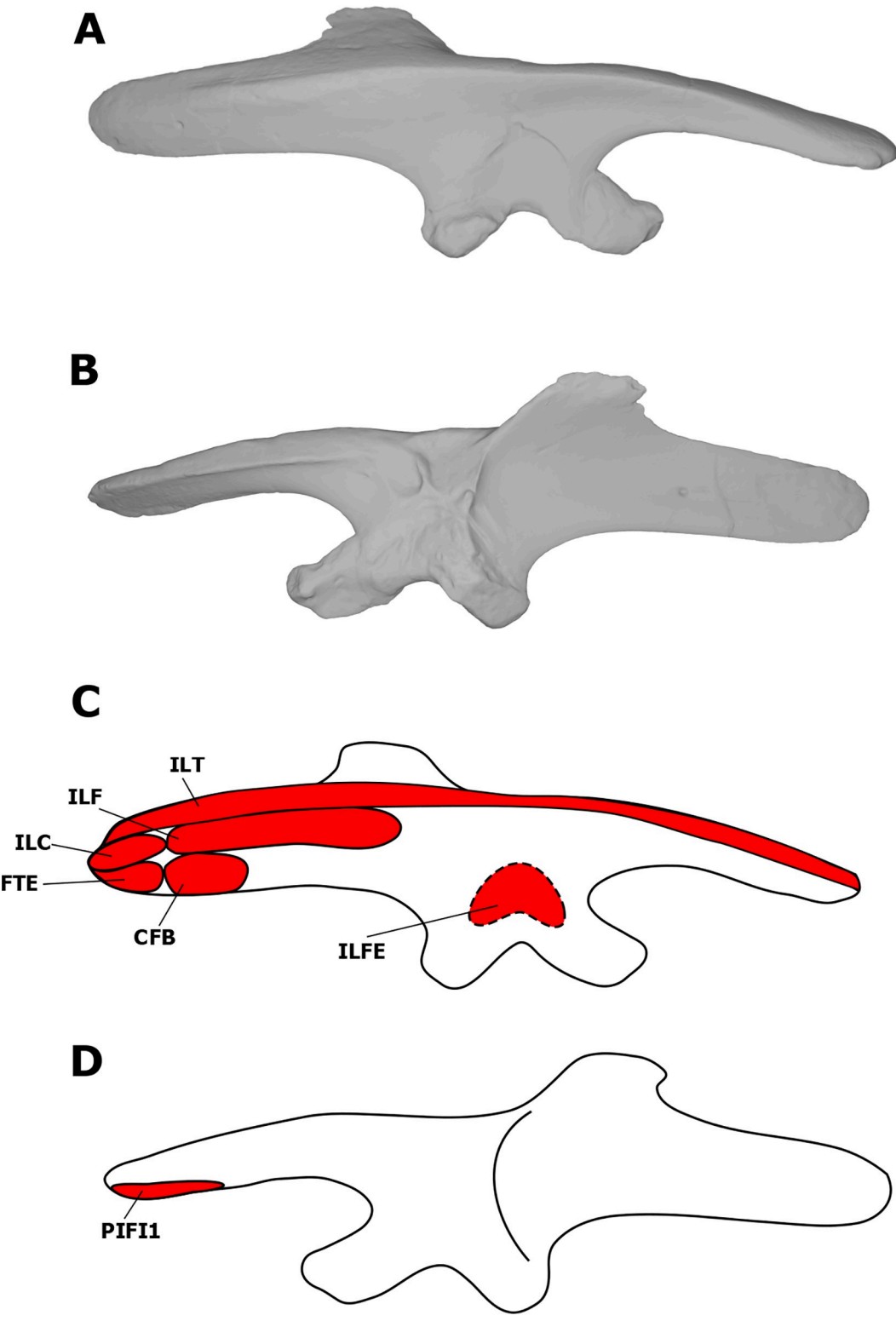

**Fig 7. Myological reconstructions and scans of the right ilium of *Stegoceras validum*.** (A) Lateral scan, (B) medial scan, (C) lateral reconstruction, and (D) medial reconstruction. Areas of red indicate muscle origins. Dashed lines indicate muscles that were reconstructed entirely based on EPB comparisons.

The muscle originates from the lateral surface of the distal shaft of the ischium in the EPB, just lateral to the attachment of the puboischiofemoralis externus III [22, 23, 77]. UALVP 2 displays distinct striations in this region of the ischium. However, attachments for the ISC and the puboischiofemoralis externus cannot be distinguished from each other. Therefore, the distal portion of this area is assigned to the origin of the ISC in *S. validum* (Fig 8).

As mentioned above, the insertions of the ISC are shared with the ILC. We therefore infer that the ISC of *S. validum* inserted on the caudal ribs and haemal arches of the anterior caudal vertebrae.

**Puboischiofemoralis internus (PIFI).** The puboischiofemoralis internus has two heads in crocodilians [22, 44]. Homologous muscles in birds are the iliofemoralis internus and the iliotrochantericus cranialis and medius [44, 78]. In crocodilians, PIFI 1 is the smaller of the two heads and originates on the medial surface of the preacetabular process of the ilium, ventral to the medial ridge and the anterior surface of the first sacral rib [44, 79]. PIFI 2 is larger and originates on the lateral faces of the dorsal vertebrae [44, 79]. In UALVP 2, the preacetabular process of the ilium is expanded and rotated medially, producing a large and broad ventromedial surface. This same condition is observed in stegosaurs (*Stegosaurus*, *Kentrosaurus*, and *Dacentrurus*), ankylosaurs (*Euoplocephalus*, *Dyoplosaurus*, and *Struthiosaurus*), and *Zalmoxes shquiperorum* Weishampel *et al.*, 2003 [80], and the resulting increase in surface area is attributed to the origin of PIFI 1 [45, 80]. The ventromedial ridge of the preacetabular process of the ilium in UALVP 2 is notably rugose, as is its ventral surface. We therefore infer that this surface served as the attachment for the PIFI 1 in *S. validum*, as it does in crocodilians (Fig 7). The dorsal vertebrae of UALVP 2 likewise preserve a rugose texture on the lateral surfaces of the centra that would have served as the attachment sites for the PIFI 2.

Some have hypothesized that the PIFI 1 inserted on the posteromedial surface of the femur near the insertion of the caudofemoralis longus (CFL) in dinosaurs [22, 44, 81]. In crocodilians, the insertion surrounds that of the CFL mediodistally and extends to the distalmost extent of the fourth trochanter [29, 44, 79–82]. There is a distinct pit medial to the fourth trochanter on the femora of UALVP 2 that likely corresponds to the insertion of the CFL. The area medial and distal to the pit is highly rugose and is attributed to the insertion of PIFI 1 in *S. validum* (Fig 9). This muscle would have abducted and extended the femur [81, 83].

PIFI 2 inserts on the anterolateral portion of the proximal femur in the EPB [22, 44, 81]. This area corresponds with the proximodistally elongate lesser trochanter of UALVP 2 (Fig 9). The surface of the trochanter is slightly rugose and is likely associated with the insertion of PIFI 2. This muscle complex would have flexed the hip [81, 83].

**Iliofemoralis (ILFE).** The origin of the iliofemoralis is consistent across the EPB and is therefore unequivocal in dinosaurs [22, 23, 44, 84–86]. In the EPB, this muscle originates from the supracetabular region of the lateral surface of the ilium and usually does not leave a scar [22, 23, 44, 84–86]. This surface is smooth in UALVP 2, but we assume that it is the origin site of the ILFE in *S. validum*, based on the EPB (Fig 9).

The insertion(s) of the ILFE are equivocal in dinosaurs [22, 44, 81]. Crocodilians possess a single-bodied ILFE that is homologous to the iliofemoralis externus (IFE) and iliotrochantericus (ITC) of birds [44, 78]. In crocodilians, the ILFE inserts on the anterolateral surface of the femur between the attachments of the femorotibialis (FMT) internus and externus [44]. The ITC of birds inserts on the proximal trochanteric crest of the femur and the IFE inserts on a ridge distolateral to the trochanteric crest, which represents the fusion of the greater and lesser trochanters [44, 82]. The only homologous feature to the attachment sites observed in the EPB on the femur of UALVP 2 is the lesser trochanter. We therefore reconstruct the ILFE here as inserting on the lesser trochanter along with the insertion of the PIFI 2 because there is no way

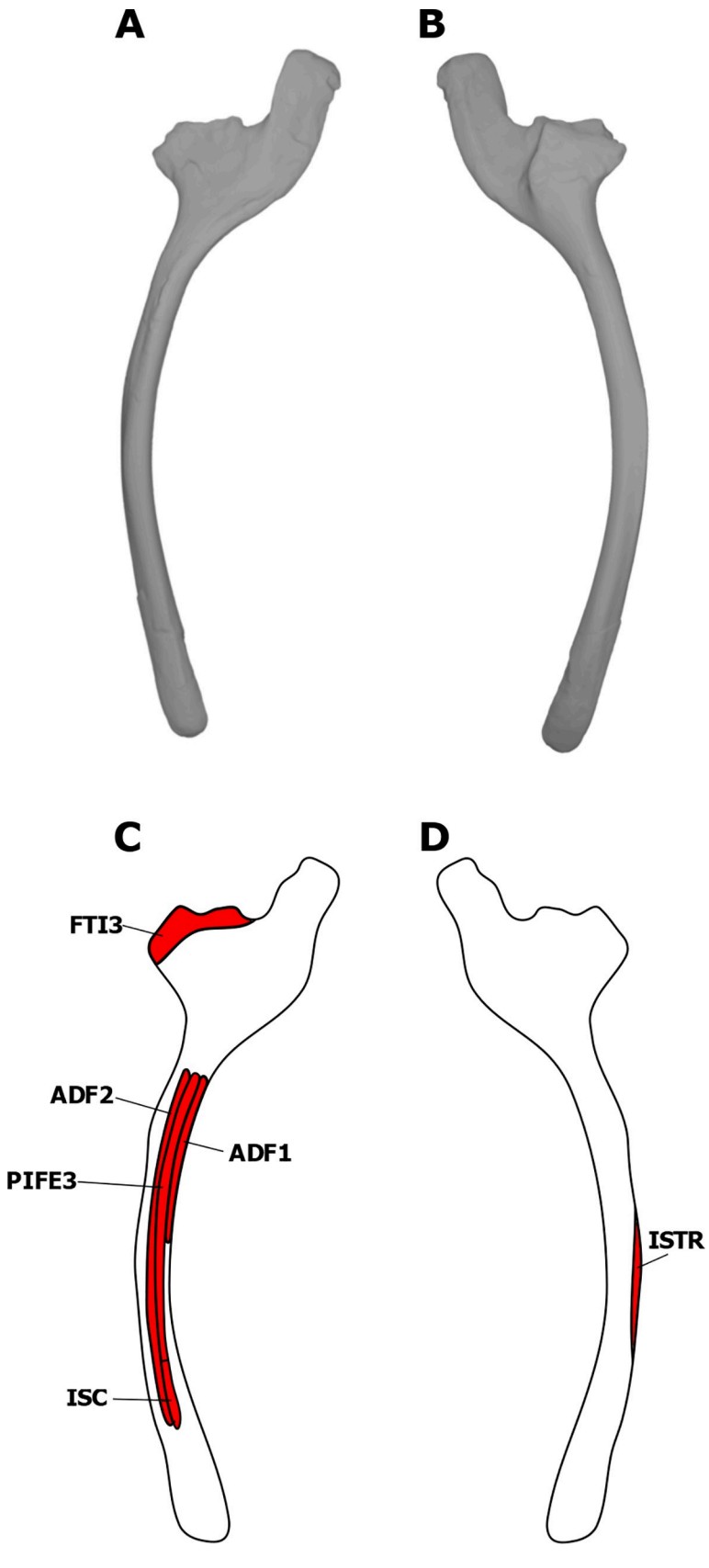

**Fig 8. Myological reconstructions and scans of the right ischium of *Stegoceras validum*.** (A) Lateral scan, (B) medial scan, (C) lateral reconstruction, and (D) medial reconstruction. Areas of red indicate muscle origins.

to differentiate scarring on the feature (Fig 9). Such a morphology would have resulted in a muscle that abducted the hip [81, 83].

**Puboischiofemoralis externus (PIFE).** The homology of the PIFE in archosaurs and its number of divisions are debated in the literature, largely because it is still not clear which muscle in birds is equivalent to the PIFE of crocodilians [44, 79, 82, 87–89]. For simplicity and consistency, the PIFE complex is assumed here to have three heads in crocodilians, two originating from the pubis and one from the ischium, and is homologous with the obturator in birds [44, 87, 88]. PIFE 1 attaches to the medial aspect of the pubis in crocodilians and PIFE 2 originates from the posterolateral pubis [44]. Birds possess a retroverted pubis, so the attachments of PIFE 1 and 2 have shifted to the postpubis [44]. The postpubis, however, is extremely reduced in basal ornithischians and pachycephalosaurs, and was unlikely able to support major muscle attachments [2, 44]. As a result, it is difficult to interpret the origins of PIFE 1 and 2 in *S. validum*, especially because UALVP 2 lacks a preserved pubis. It has been hypothesized that the pubic attachments of PIFE 1 and 2 shifted to the ischiadic apron in ornithischians [44, 63, 90, 91]; however, this structure is also heavily reduced in ornithischians and would result in an overall reduction of the PIFE musculature. There is also no osteological correlate present on this region of the ischium in UALVP 2. Alternatively, the first and second heads of the PIFE complex may have been completely lost in ornithischians [45]. Because there is no preserved pubis and no osteological correlate for PIFE 1 and 2 in UALVP 2, these attachments are not reconstructed here for *S. validum*. It is, however, still possible that these attachments were present in the animal.

The PIFE 3 of crocodilians originates from the lateral part of the ischium and has no avian homologue [44]. Romer (1927) [88] reported, based on the embryonic development of birds, that the lack of a third head of the obturator complex was a derived character of the clade. Maidment & Barrett (2011) reconstructed the pelvic musculature of basal ornithischians as having a crocodilian morphology of the PIFE complex with all three heads. Similarly, Voegele *et al.* (2021) reconstructed the pelvis of *D. schrani* as having a PIFE 3 on the ventrolateral face of the distal shaft of the ischium, based on distinct striations in that region. UALVP 2 has a notably rugose ridge on the lateral surface of the ischium that descends distally from the base of the pubic peduncle, approximately halfway down the shaft. This ridge corresponds to the origin of the PIFE 3 in crocodilians and the previously-mentioned dinosaurian muscle reconstructions. It is therefore presented here as the origin site for the PIFE 3 in *S. validum* (Fig 8).

All heads of the PIFE share one insertion in the EPB on the greater trochanter [22, 44]. The greater trochanter of the right femur of UALVP 2 displays rugose scarring on its posterolateral surface, which is attributed here to the insertion of the PIFE complex (Fig 9). Such a morphology would have resulted in a complex that flexed and adducted the hip [81, 83].

**Iliotibialis (ILT).** The presence of the iliotibialis is unequivocal in dinosaurs, based on EPB comparisons [22, 23, 44, 81, 84–86]. It attaches on the dorsolateral margin of the ilium in the EPB [22, 44, 81]. Such an attachment has also been hypothesized in ornithischians [44, 45, 80] and sauropods [22, 38, 86, 92]. In pachycephalosaurs, the dorsolateral margin of the ilium is folded medially, similar to the condition observed in *Z. shquiperorum* [80]. The entirety of this surface in UALVP 2 is covered in prominent parallel striations, supporting this as the attachment site for the ILT in *S. validum*.

The ILT consistently shares a tendinous insertion with the femorotibialis complex and the ambiens on the anteromedial surface of the cnemial crest in the EPB [22, 44, 79, 81]. This

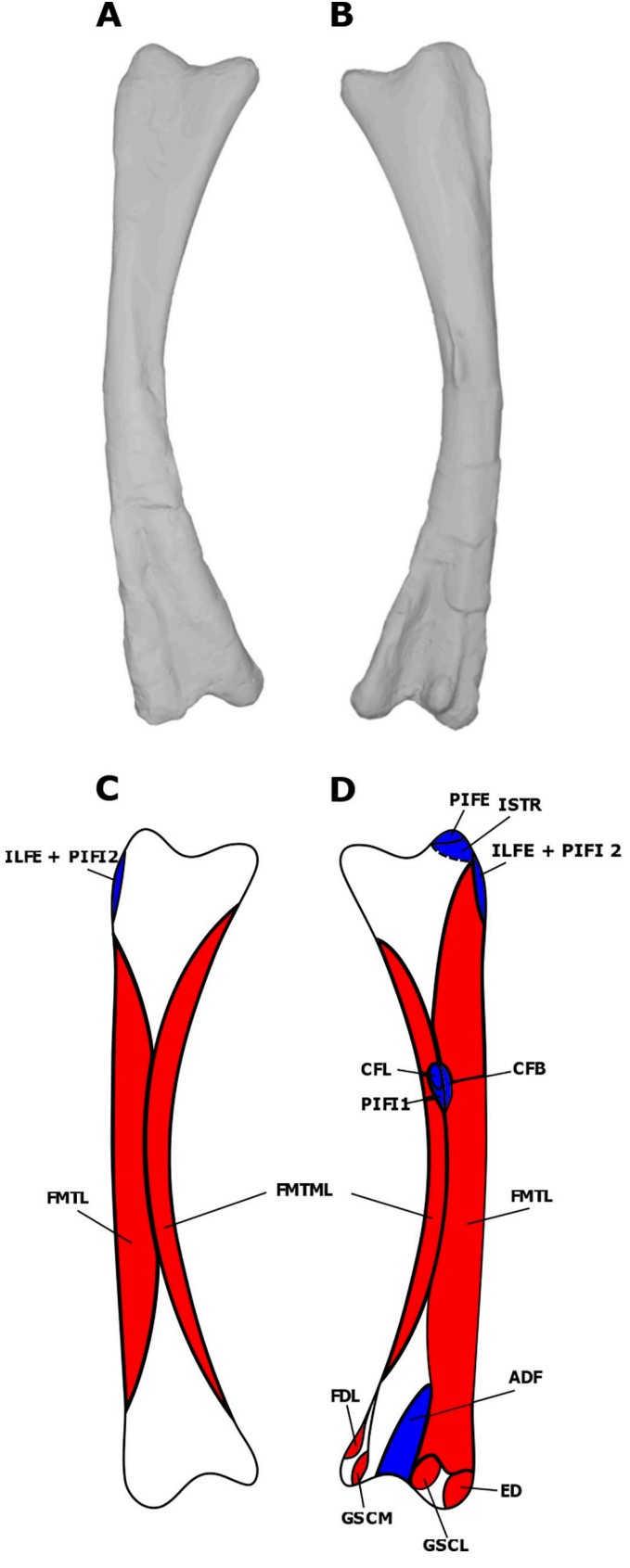

**Fig 9. Myological reconstructions and scans of the right femur of *Stegoceras validum*.** (A) Anterior scan, (B) posterior scan, (C) anterior reconstruction, and (D) posterior reconstruction. Areas of red indicate muscle origins. Areas of blue indicate muscle insertions. Dashed lines indicate muscles that were reconstructed entirely based on EPB comparisons.

region of the tibia is densely striated in UALVP 2 and is therefore attributed to the insertion of this tendon in *S. validum* (Fig 10). This muscle flexed, extended, and abducted the hip [83].

Continuous striations similar to those associated with the origin of the ILT extend onto the dorsal surface of the medial iliac flange in UALVP 2. No other muscles attach to the corresponding area in extant archosaurs [22, 23, 44, 81, 84, 85], making it difficult to assess what may have left these scars on the ilium of UALVP 2. It is possible that this area may have served as an attachment site for sacroiliac ligaments or perhaps for a muscle/belly unique to pachycephalosaurs. Because the EPB and no other dinosaurs display such a structure, no muscles can be confidently reconstructed as attaching here in *S. validum*. We reconstruct the ILT with an origin covering the entire length of the dorsal margin of the ilium.

**Iliofibularis (ILF).**   The presence and structure of the iliofibularis is consistent across the EPB [22, 23, 44, 84–86]. It attaches to the dorsolateral margin of the postacetabular process of the ilium in extant archosaurs [22, 44, 81]. This same morphology has been reconstructed over a wide variety of dinosaur taxa [22, 23, 44, 45, 80, 81, 85, 92]. The entire length of the dorsolateral margin of the postacetabular process of UALVP 2 displays parallel linear striations that likely correspond to the origin of the ILF. As such, the ILF is reconstructed here as attaching to the lateral surface of the postacetabular process, anterior to the ILC, in *S. validum* (Fig 7).

The insertion site of the ILF occurs on the lateral aspect of the proximal fibula in archosaurs [22, 23, 44, 81, 83–86, 93]. The anterolateral surface of the left fibula of UALVP 2 is heavily rugose and is attributed to this insertion in *S. validum* (Fig 11). This muscle would have abducted the hip [81, 83].

**Flexor tibialis internus (FTI).**   The flexor tibialis internus is present in EPB taxa but the number of heads varies from four in crocodilians to just one in birds [44]. FTI 1 is present in crocodiles, but appears to have been lost in birds [44, 82]. FTI 2 is only present in crocodilians and has no avian homologue, and so it is likely that this represents a crocodilian autapomorphy [44, 82]. FTI 3 and 4 of crocodilians are suggested to be homologous to the flexor cruris medialis in birds and the FTI 2 of turtles and lepidosaurs [44, 82]. FTI 4, however, has no osteological attachments and so cannot be confirmed in fossil taxa [44, 79]. Given the confusing phylogenetic origins of the FTI heads, only one head homologous to the FTI 3 of crocodilians will be considered here, because it is present in all consulted modern taxa. This head originates on the proximolateral ischium in crocodilians and on the distal end of the ischium in birds [29, 44, 94]. It has been proposed that the FTI 3 of basal archosaurs originated on the ischial tuberosity near its attachment in crocodilians [44, 82], a direct osteological correlate on the lateral ischium, just distal to the iliac peduncle. This area displays a heavy rugose texture in UALVP 2 and is inferred as the origin for the FTI 3 in *S. validum* (Fig 8).

The FTI musculature shares a tendinous insertion on the proximal posteromedial surface of the tibia with the flexor tibialis externus in the EPB [22, 81, 83]. This area displays a mildly striated texture on the tibia of UALVP 2, which is attributed here to the attachment of this tendon in *S. validum* (Fig 10). With such a morphology, the FTI musculature would have adducted and extended the hip, and flexed the knee [81, 83].

**Flexor tibialis externus (FTE).**   The attachment sites of the FTE are unequivocal in dinosaurs, based on the EPB. It originates on the posterior margin of the lateral face of the postacetabular process in these animals and has been reconstructed as such in many dinosaurs [22, 23, 44, 45, 81, 84, 85]. As stated previously, the lateral surface of the postacetabular process of the

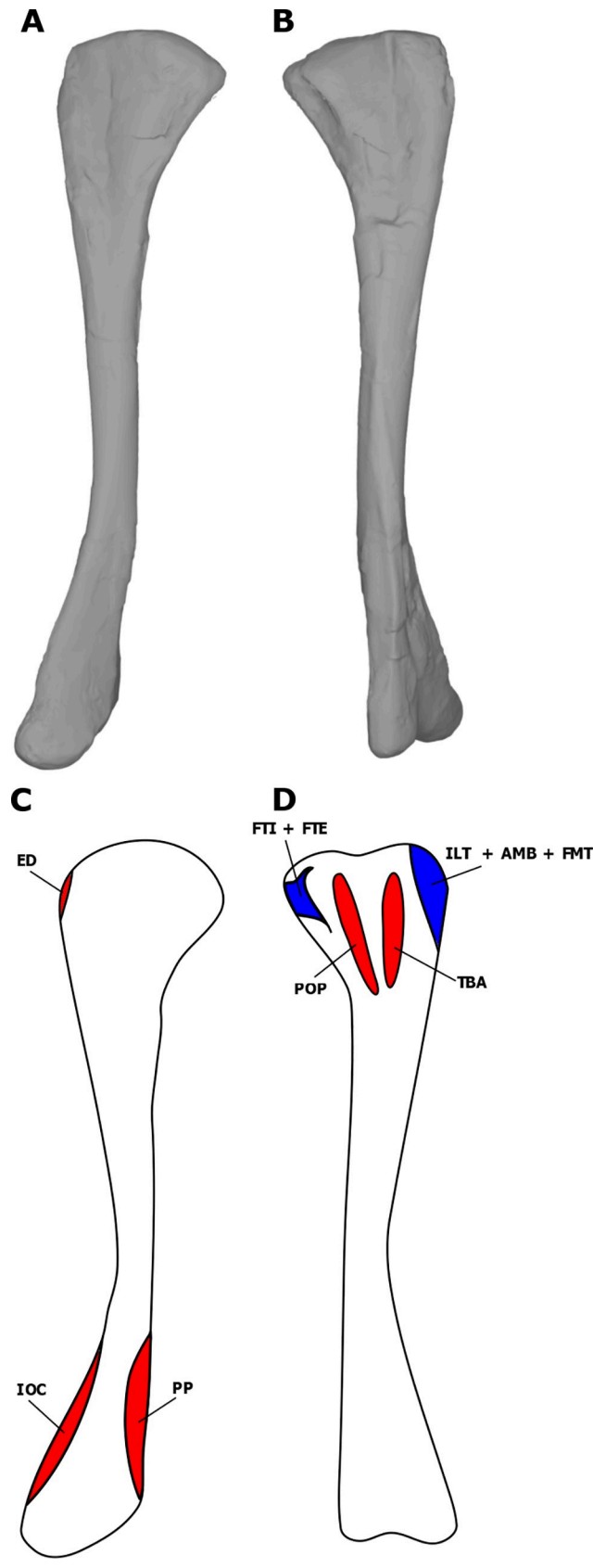

**Fig 10. Myological reconstructions and scans of the left tibia of *Stegoceras validum*.** (A) Lateral scan, (B) medial scan, (C) lateral reconstruction, and (D) medial reconstruction. Areas of red indicate muscle origins. Areas of blue indicate muscle insertions.

ilium in UALVP 2 displays prominent striations over its entire length. There is no obvious way of discerning exactly which subset of these striations is associated with the FTE versus the ILF or ILC, so the FTE is reconstructed here posterior and ventral to the ILF, as it is in crocodilians and *Silesaurus opolensis* (Fig 7) [29, 79, 81, 83, 94].

As discussed already, the FTE would have shared a tendinous insertion with the FTI musculature on the proximal posteromedial surface of the tibia in *S. validum* (Fig 10). This muscle would have flexed the knee and extended and adducted the hip along with the FTI musculature [81, 83].

**Caudofemoralis brevis (CFB).**   In crocodilians and birds, the CFB originates on the ventral aspect of the lateral surface of the postacetabular blade [29, 44, 79, 94]. This area of the ilium of UALVP 2 shows faint striations and is attributed to the attachment of the CFB, ventral to the ILF and anterior to the FTE (Fig 7).

The CFB of both EPB taxa inserts on the lateral surface of the fourth trochanter [22, 44, 81]. There is no distinct texturing on this surface in UALVP 2. However, because the fourth trochanter is a direct osteological correlate to the CFB in archosaurs, it likely still served as its insertion site in *S. validum* (Fig 9). This muscle would have extended and adducted the hip [81, 83].

**Caudofemoralis longus (CFL).**   The CFL inserts in a rugose pit medial to the proximal end of the fourth trochanter in the EPB [22, 29, 44, 79, 81]. This feature is present in UALVP 2, and so the CFL is reconstructed as inserting in this area (Fig 9). With such a morphology the CFL would have extended and adducted the hip along with the CFB [81, 83]. It would also contribute to mediolateral movement of the tail.

**Adductores femores (ADF).**   Two adductores femores are consistently present in the EPB, although their origin locations are variable [22, 23, 44, 84, 86]. In crocodilians, ADF 1 originates on the dorsolateral surface of the ischial shaft and ADF 2 originates on the ventrolateral surface of the ischium, surrounding the PIFE 3 and ISC [44, 79]. In birds, origins are found on the lateral ischium, anterior pubis, and ilioischiadic membrane [29, 44, 94]. Because birds lack a PIFE 3, the origins of the ADF 1 and ADF 2 occur adjacent to each other on the ischium [44, 82]. In UALVP 2, minor striations occur dorsal and ventral to the ridge identified as the origin of the PIFE 3. These are attributed to the origins of ADF 1 and ADF 2, based on the condition observed in the EPB (Fig 8).

ADF 1 and 2 insert together on the posterior surface of the femoral shaft in the EPB [44, 79, 81], although their exact insertion is variable. In ornithischian dinosaurs, such as *Barilium dawsoni*, *Dysalotosaurus lettowvorbecki*, *Hypsilophodon foxii*, *Kentrosaurus aethiopicus*, and *Mantellisaurus atherfieldensis*, there are clear muscle scars along the linea intermuscularis caudalis dorsal to the lateral femoral condyle which are attributed to the ADF musculature [44, 45]. Piechowski & Talanda (2020) place this attachment slightly more medial between the lateral and distal condyles, based on scarring observed in the dinosauriform *Silesaurus opolensis*. In UALVP 2, the observed condition is similar to that of *S. opolensis*. There are distinct striations between the distal condyles of the left femur that extend proximally up to roughly one third of the femoral shaft. The ADF musculature of *S. validum* is therefore assumed to have inserted here in the same area proposed by Piechowski & Talanda (2020) (Fig 9). These muscles would have adducted and extended the hip [81, 83].

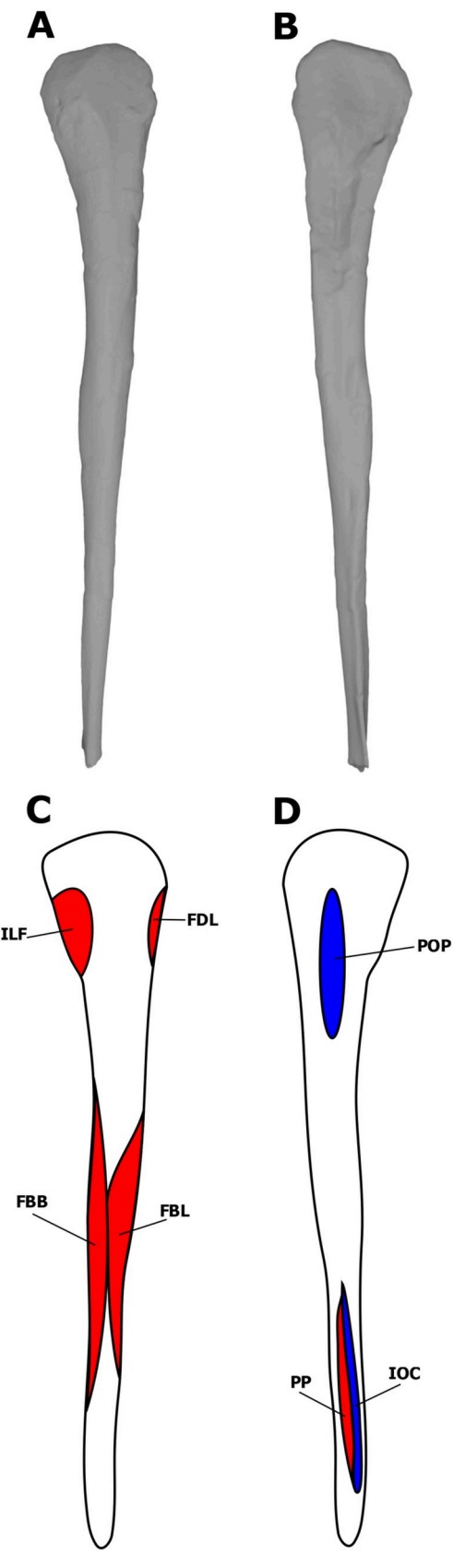

**Fig 11. Myological reconstructions of the left fibula of *Stegoceras validum*.** (A) Lateral scan, (B) medial scan, (C) lateral reconstruction, and (D) medial reconstruction. Areas of red indicate muscle origins. Areas of blue indicate muscle insertions.

## Ischiotrochantericus (ISTR)

The ischiotrochantericus originates on the medial ischial shaft in crocodilians and on the lateral surface of the ischial shaft in birds [23, 44, 79]. It is likely that the crocodilian condition is ancestral, and that the lateral migration of the ISTR in birds is a result of the loss of PIFE 3, which attaches to the lateral ischial shaft [44, 64, 88]. There are no obvious scar textures preserved on the medial ischial shaft of UALVP 2; however, there is a smooth, distinct ridge on its dorsomedial surface. A similar ridge is reported in *D. schrani* [22] and is recognized as the origin of the ISTR. This ridge is also attributed here to the origin site of the ISTR in *S. validum* (Fig 8).

The ISTR of crocodilians inserts on the proximolateral face of the femur, just distal to the insertion of the PIFE complex [44, 79]. In birds, it inserts on the trochanteric crest of the femur [44, 82]. In dinosaurs and basal archosaurs, this attachment may be found on either the proximal posterolateral surface of the femur, as in crocodiles, or on the greater and lesser trochanters, which are homologous to the trochanteric crest of birds. Both conditions have been previously reconstructed [22, 44, 81, 86]. Without any definitive osteological correlates in these regions of the femora of UALVP 2, the insertion of the ISTR is tentatively reconstructed here just distal to the insertion of the PIFE complex, as in crocodilians (Fig 9). This muscle would have supinated and retracted the hip [81, 83].

## Ambiens (AMB)

The ambiens is unequivocal in archosaurs but its number of divisions varies in the EPB. Crocodilians possess a two-headed AMB, a unique condition among modern reptiles, and birds and other reptiles display a single head [22, 29, 44, 79, 82, 88]. In crocodilians, the muscle originates proximomedially on the pubis; it originates anteriorly in birds [29, 44, 79]. Because a two-headed AMB is an autapomorphy of crocodilians, it is likely that dinosaurs had a single-headed AMB. It is thought that basal archosaurs would have had an AMB attachment on the proximoanterior pubic tubercle [44, 82, 89, 95], a direct osteological correlate for this muscle. However, because ornithischians display a retroverted pubis, the pubic tubercle is located ventrally on the pubis [44, 82]. Given the drastic change of this attachment site, it is unlikely that the AMB still attached to it in ornithischians, such as *S. validum*. It has been hypothesized that the AMB instead arises from the prepubis in ornithischians, which is supported by osteological correlates found in *Heterodontosaurus tucki*, *Scelidosaurus harrisonii* Owen, 1859, *Edmontosaurus regalis* Lambe, 1917, *Gilmoreosaurus mongoliensis* Brett-Surman, 1979, and *Hypsilophodon foxii* [44, 45, 94, 96]. Although the pubis of UALVP 2 is not preserved, it is unlikely that the AMB still attached to the pubic tubercle, due to the retroversion of the pubis and its extreme reduction in other pachycephalosaurs [2]. As such, the AMB of *S. validum* is tentatively reconstructed here as originating from the proximolateral prepubis, as it did in some other ornithischians. Given that the pubis and prepubis of pachycephalosaurs are both highly reduced [2], it is likely that the muscles that attached to them, such as the AMB, would have been reduced as well. The AMB would have shared a tendinous insertion with the femorotibialis and ILT musculature, as described previously (Fig 10).

## Hind limb musculature

**Femorotibiales (FMT).** The presence of the femorotibiales musculature is invariant in extant archosaurs; however, the number of heads differs in the EPB [44, 79, 81]. Crocodilians

possess two heads: the femorotibialis externus (FMTE) and the femorotibialis internus (FMTI) [44, 79]. The FMTE originates on the posteromedial femoral shaft between the adductors and fourth trochanter, and the FMTI originates over most of the anterior and medial femoral shaft. Birds possess an additional head: the femorotibialis medius (FMTM) [44, 79]. The avian FMTE originates on the posterolateral femoral shaft, the FMTI originates on the posteroventral aspect of the shaft, and the FMTM originates on the anterior and medial surfaces of the shaft [44]. In both birds and crocodilians, these muscles are clearly separated from each other by intermuscular lines [44, 81]. The femora of UALVP 2 display an intermuscular line on their posterior surfaces, extending proximally and distally from the fourth trochanters, which would have mediolaterally divided the FMT musculature. A ridge dividing the medial and lateral aspects of the anterior shaft is also present. As such, there were probably only two heads of the FMT in *S. validum*: the femorotibialis lateralis (FMTL), which occupied the lateral surfaces of the anterior and posterior parts of the femoral shaft, and the femorotibialis medialis (FMTML), which occupied the medial surfaces of the anterior and posterior regions of the femoral shaft (Fig 9). As discussed already, the FMT complex would have shared a single tendinous insertion with the AMB and ILT on the cnemial crest of *S. validum* (Fig 10).

**Gastrocnemius (GSC).** The EPB displays at least two divisions of the gastrocnemius: the gastrocnemius lateralis (GSCL), which originates from the posteromedial aspect of the lateral condyle of the femur, and the gastrocnemius medialis (GSCM), which originates from the medial surface of the proximal tibia [22, 81–83]. Birds possess a third division: the gastrocnemius intermedialis (GSCI), which originates on the posteromedial aspect of the medial femoral condyle [81]. Both femora of UALVP 2 display highly rugose texturing on their distal condyles, supporting the presence of the GSCL and the GSCI. Their attachments are proposed to originate on the posteromedial aspects of the femoral condyles in *S. validum* (Fig 9).

The GSC musculature shares common tendinous insertions on the ventral surfaces of metatarsals II-IV in the EPB [81, 83, 94, 97–99]. These elements are preserved in UALVP 2 and display rugose texturing on their proximoventral surfaces. This scarring is attributed here to the insertions of the GSC tendons in *S. validum* (Fig 12). With such a morphology, the GSC would have extended the ankle and flexed the knee [81, 83].

**Fibularis (FB).** The presence of the fibularis is unequivocal in dinosaurs, although its area of origin is equivocal. Crocodilians possess both a fibularis longus (FBL) and a fibularis brevis (FBB) attaching to the lateral fibular shaft, with the former posterior to the latter. Because birds have a highly reduced and fused fibula, the FBL has migrated and attaches to the cristae of the tibia [81, 83]. The tibia possesses no such structures in UALVP 2, and the fibula is not heavily reduced, and therefore it is likely that *S. validum* displayed the crocodilian origins of the FB musculature. Additionally, faint striations are visible on the antero- and posterolateral surfaces of the left fibular shaft of UALVP 2, which likely correspond to the origins of the FBB and FBL, respectively (Fig 11).

These muscles insert on the ventral part of the calcaneum and distoventral region of metatarsal V in the EPB [81]. Neither of these elements are preserved in UALVP 2, so the FBL is tentatively reconstructed here as attaching to the ventral part of the calcaneum, whereas the FBB is reconstructed as attaching to the distoventral surface of metatarsal V, based on EPB comparisons. These muscles would have flexed the ankle [81, 83].

**Extensor digitorum (ED).** The origin of the extensor digitorum longus (EDL) is equivocal in dinosaurs, based on the EPB [22, 23, 81, 84, 85]. In crocodilians, the origin is found on the lateral aspect of the lateral distal condyle of the femur [22, 23, 84]. In birds, it is found on the anterior surface of the cnemial crest [22, 23, 84]. The lateral condyles of the femora of UALVP 2 both display heavy, rugose texturing on their lateral surfaces. However, the cnemial crest of the tibia also displays a striated texture on its anterior surface. There is no way of

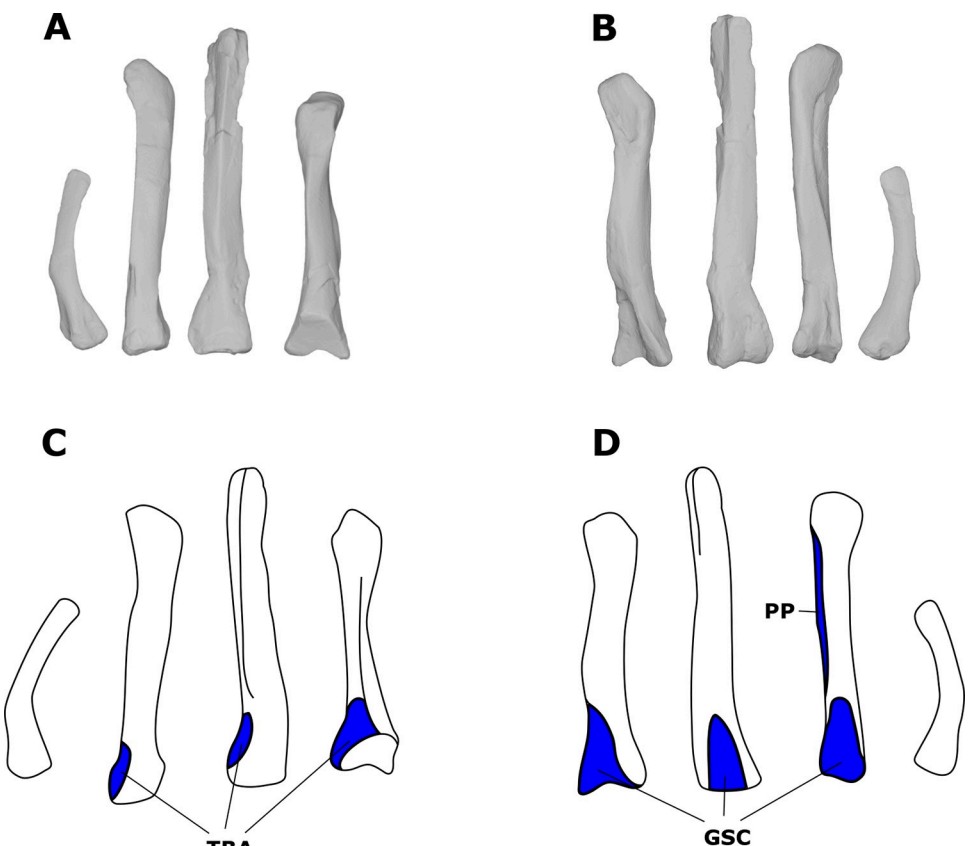

**Fig 12. Myological reconstructions and scans of the left Metatarsals I-IV of *Stegoceras validum*.** (A) Dorsal scan, (B) ventral scan, (C) dorsal reconstructions, and (D) ventral reconstruction. Areas of blue indicate muscle insertions.

distinguishing which of these osteological correlates corresponds to the origin of the EDL. As such, the origin is tentatively reconstructed here as originating from both areas, as in the reconstructions of Dilkes (2000) [23], Carrano & Hutchinson (2002) [84], and Voegele *et al.* (2021) [22] (Figs 9 & 10). It is most likely that the EDL would have had only one of these origins, but without more data, it is impossible to determine which is more likely. The extensor digitorum brevis (EDB) originates from the dorsal surfaces of the proximal tarsals in crocodilians and is fused with the EDL in birds [81, 100]. Because the tarsals are not preserved in UALVP 2, it is tentatively reconstructed as originating from the same location as crocodilians in *S. validum*, because birds possess a highly modified tibiotarsus and pes.

Whether these muscles shared insertions on the dorsal unguals and phalanges or had individual insertions in *S. validum* is impossible to know, because these elements are not preserved in UALVP 2. However, it is likely that these muscles both inserted somewhere on these elements because they do in both crocodilians and birds [65, 81, 83, 84, 97–99, 101]. With such morphologies, both muscles would have extended the digits of the pes [81, 83].

**Flexor digitorum longus (FDL).**   The origins of the flexor digitorum longus are equivocal in dinosaurs, based on the EPB. Osteological correlates for this muscle are consistently found on the posterolateral aspect of the lateral femoral condyle [22, 23, 30, 81, 84, 85]. The femora of UALVP 2 also display distinct rugose scarring on the lateral aspects of the lateral condyles. *Stegoceras validum* is therefore reconstructed as having an origin for the FDL in this location (Fig 9). The fibular origin of the FDL is lost in birds as a result of their highly modified

tibiotarsus. In crocodilians, this origin is found on the posterolateral surface of the proximal fibula [81]. There is a striated texture on this region of the left fibula of UALVP 2 that likely corresponds to the origin of the FDL. As such, *S. validum* is reconstructed here as possessing a fibular head of the FDL originating from the proximal posteromedial surface of the fibula (Fig 7).

The insertion of the FDL occurs on the ventral unguals and phalanges II-IV in the EPB [23, 81, 83, 94, 98, 99, 101, 102]. Because these elements are not present in UALVP 2, FDL insertions are assumed to be the same in *S. validum* as they are in the EPB. This muscle would have extended the ankle and flex the pedal digits [81, 83].

**Popliteus (POP).** The origin of the popliteus on the tibia is somewhat variable in the EPB, but the muscle generally originates from the proximal lateral surface of the cnemial pocket [22, 81, 84]. This area also gives rise to the tibialis anterior, which is generally found anterior to the POP attachment [81]. UALVP 2 displays subtle striations across this entire region. As such, striations on the anterior side of the cnemial pocket are attributed to the origin of the tibialis anterior whereas the posterior striations are attributed to the POP (Fig 10).

The insertion of the POP is found on the proximal medial surface of the fibula in archosaurs and is often denoted by a scarred concavity [58, 81]. The left fibula of UALVP 2 has been taphonomically flattened in the transverse plane and displays no such concavity, but it is present in the right fibula. As such, the POP is reconstructed here as having a fibular insertion in this cavity of the fibula in *S. validum* (Fig 11). This muscle would have rotated the fibula about the length of the tibia [81, 83].

**Tibialis anterior (TBA).** As previously discussed, the TBA originates from the anterior aspect of the cnemial pocket of the tibia in *S. validum*, based on the condition observed in the EPB and osteological correlates on the tibia of UALVP 2 (Fig 10).

The TBA inserts on the proximolateral surfaces of metatarsals II-IV in the EPB [23, 81, 83, 94, 98, 99]. Faint striations are present on these surfaces of the metatarsals of UALVP 2. As such, the TBA is reconstructed here as inserting on the proximolateral surfaces of metatarsals II-IV in *S. validum* (Fig 8). This muscle would have flexed the ankle [81, 83].

**Interosseus cruris (IOC).** The presence of the IOC is unequivocal in dinosaurs, based on the EPB. The muscle originates from the lateral tibia and attaches to the distomedial fibula [65, 81, 84, 97, 100, 102]. The tibia of UALVP 2 displays striations on its distal anterolateral surface, which are attributed here to the tibial origin of the IOC in *S. validum* (Fig 10). Distinct rugose ridges also occur on the distomedial fibular surfaces of UALVP 2, which likely correspond to the insertion of the IOC (Fig 11). This muscle would have flexed the ankle [81].

**Pronator profundus (PP).** The pronator profundus is unequivocally present in dinosaurs, but its number of divisions is equivocal [81, 97, 100, 101]. Crocodilians possess both a tibial and fibular head; birds possess only a tibial head because the fibula has been drastically reduced and fused to the tibiotarsus [81, 97, 100, 101]. In crocodilians, the tibial head of the PP originates from the distal half of the posterolateral surface of the tibia [81]. There are notable striations on this surface in UALVP 2, which are attributed here to the origin of the PP (Fig 10). The fibular origin of the PP is located on the distal posteromedial surface of the fibular shaft posterior to the IOC in crocodilians, and is often marked by scarring or a small ridge [81, 97, 100, 101]. Such a ridge is present on both fibulae of UALVP 2, but it has slightly shifted to the anteromedial surface of each fibula. The posterior aspect of this ridge is attributed here to the fibular origin of the PP in *S. validum* (Fig 11).

The PP inserts on the ventromedial surface of metatarsals I and II in extant archosaurs [81, 97, 102]. Metatarsal I of UALVP 2 is somewhat reduced and displays no osteological correlates of the PP. Metatarsal II, however, displays a rugose texture on its ventromedial surface that is

attributed here to the insertion of the PP in *S. validum* (Fig 12). Such a morphology would have resulted in a muscle that flexed the ankle [81].

By combining the previous individual muscle reconstructions of the pelvis and hind limb muscles, we created a 3D model of the pelvic and hind limb musculature of *S. validum* (Fig 13). The model shows the origin(s) and insertion(s) of each muscle and how the muscles would have interacted in three dimensions.

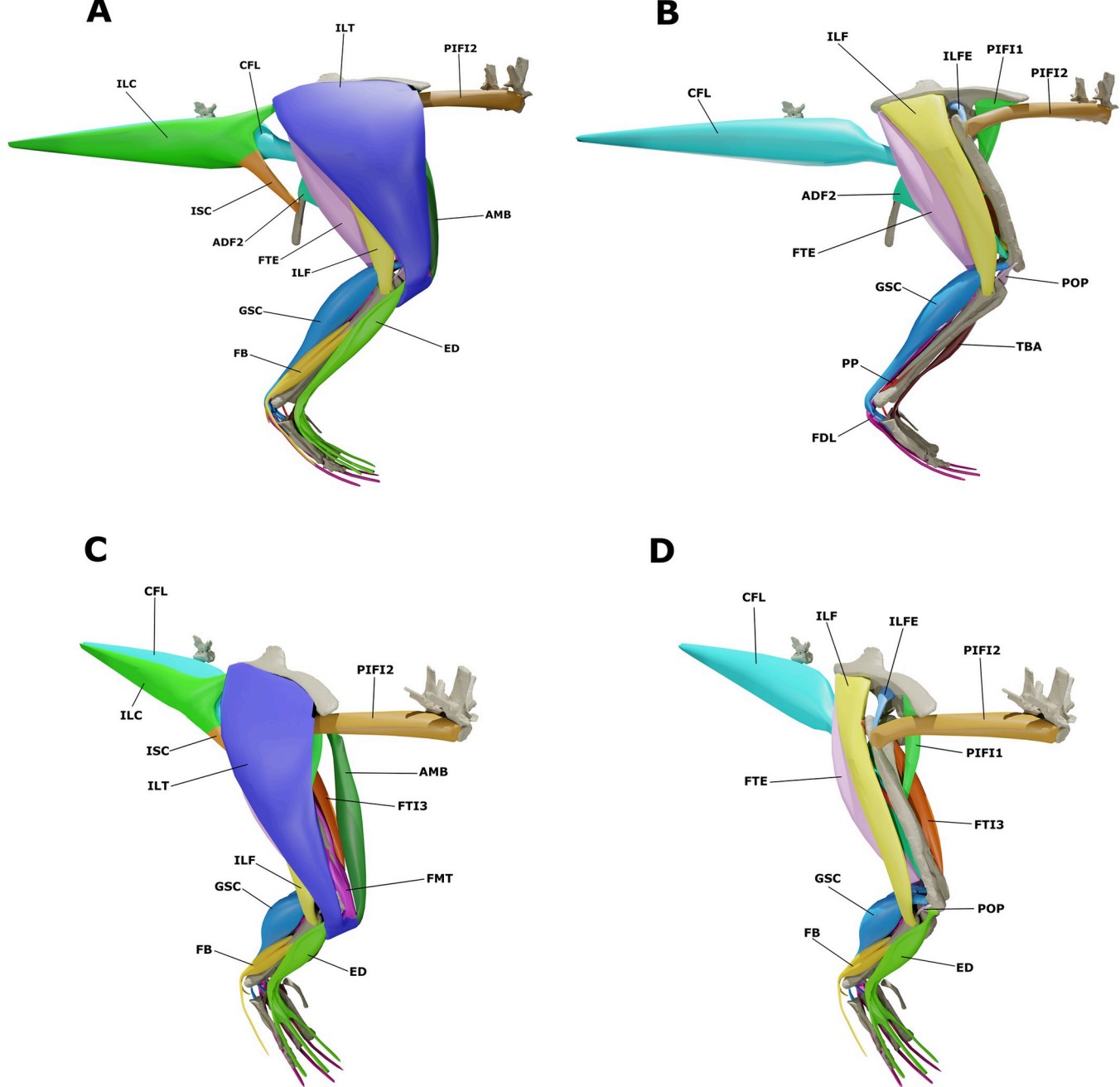

**Fig 13. Pelvic and hind limb muscular reconstruction of *Stegoceras validum*.** (A) Superficial musculature in lateral view. (B) Deep musculature in lateral view. (C) Superficial musculature in anterodorsolateral view. (D) Deep musculature in anterodorsolateral view.

## Discussion

### Paleobiology of the pachycephalosaur forelimb

The pectoral and forelimb myology *of S. validum* is generally conservative, although probably less so than reported here due to our conservative methodologies. Most appendicular muscles in the Extant Phylogenetic Bracket of dinosaurs have clear osteological correlates at the expected areas in UALVP 2. Deviations from the conditions observed in the EBP are easily accounted for by large scale locomotory differences between the crocodilians and birds that constitute the EPB of the non-avian dinosaurs. For example, the bipedal stance of pachycephalosaurs means that their forelimbs did not need to support their body weight. Because crocodilians are quadrupedal animals, their forelimb musculature must be strong enough to support their body weight during terrestrial locomotion. As a result, many of their pectoral and forelimb muscles, and their corresponding osteological correlates, are well-developed compared to the inferred musculature of *Stegoceras validum*. The avian pectoral girdle and forelimb have been highly modified for flight in most taxa and possess modifications including fused carpals and metacarpals into the carpometacarpus, fused clavicles into a single furculum, and an enlarged sternum with a well-developed ventral keel [103]. Pachycephalosaurs possess no such modifications to their forelimb osteology, and therefore do not display any of the corresponding avian myological conditions [31, 35, 37, 44, 79].

Although pachycephalosaurs are widely regarded as herbivores [2, 104, 105], the recent report of laterally compressed, serrated anterior dentary teeth in young *Pachycephalosaurus wyomingensis* [104] might lead some to wonder whether these animals tended towards carnivory. To this end, the existence of some key differences are noted between the forelimb musculature of *S. validum* and carnivorous or omnivorous bipedal, early saurischians. First, early-diverging saurischians such as *Thecodontosaurus antiquus*, *Guaibasaurus candelariensis*, and *Saturnalia tupiniquim* exhibit a larger deltopectoral crest than pachycephalosaurs and bipedal ornithischians more generally [45], resulting in a longer moment arm for the SC and a larger insertion area for the DCL [35, 54]. Second, basal theropods also display a longer scapular blade, providing a more distal attachment area for the DSC [35]. Finally, basal theropods possess a more distal insertion of the latissimus dorsi on the humerus. These morphologies result in more mechanically efficient muscles functioning in humeral extension in early-diverging saurischians [35], and almost certainly had some biological role in the subduing and killing of prey. These adaptations are altogether lacking in *S. validum*; the reconstructed forelimb musculature of *S. validum* compares more closely with those of other bipedal, herbivorous ornithischians [44], and likely functioned in grasping and manipulating small food items. Some bipedal ornithischians, however, have been suggested to have had additional uses and adaptations of the forelimb. One such example is *Oryctodromeus cubicularis* Varricchio *et al*., 2007 [106]. It has been suggested that *O. cubicularis* engaged in burrowing behaviours on the basis of forelimb modifications including a pronounced deltopectoral crest of the humerus, prominent acromial spine of the scapula, and greatly expanded posterior scapular blade [106]. These are osteological correlates of the deltoideus scapularis, deltoideus clavicularis, and the teres major, which are all muscles associated with digging in modern burrowing taxa [106]. The pronounced osteological correlates of these muscles in *O. cubicularis* would have resulted in broader muscle attachments and stronger muscles which would have been beneficial for burrowing [106]. No such adaptations are present in UALVP 2, and so it is unlikely that *S. validum* would have partaken in such specialized behaviours.

## Postcranial adaptations for agonistic combat

Pachycephalosaurs exhibit several odd modifications of the pelvic and hind limb skeleton that are reasonably interpreted as adaptations for intraspecific head- or flank-butting behaviour [13–15, 18]:

1. The corrugated zygapophyses of the dorsal vertebrae would have further added to the rigidity to the axial column by greatly limiting lateral flexion of the trunk [107]. This may have allowed pachycephalosaurs to maintain a straight axial posture when head-butting, limiting injuries caused by accidental flexion or buckling of the vertebral column.

2. The pelvis is broad and the hind limbs are comparatively stout, having a tibia that is shorter than the femur [45]. These traits may have served to both broaden the stance and lower the centre of gravity of the body, which in turn would have provided added stability during head-butting contests.

3. Pachycephalosaurs are the only tetrapods known to possess a caudal basket of myorhabdoi [16]. These structures are thick, rigid, and extend across the first five to six caudal vertebrae. As such, they would have added significant rigidity to the base of the tail. It has been suggested that the stability provided by the myorhabdoi would have facilitated a possible "tripodal prop" during head-butting competitions [16, 107]. A second hypothesis suggests that the ossified myosepta may have acted as armour against flank-butting [108]. However, this hypothesis has been criticized because significant gaps exist between adjacent segmental complexes. This pattern does not match the better-known superficial armour of other archosaurs such as ankylosaurs or crocodilians [16].

The associated musculature of the pelvis and hind limb also displays unique morphologies that we interpret as adaptations for head-butting:

1. The characteristic medial iliac flange is striated dorsally, but probably did not evolve to enlarge the surface area for the origin of the ILT, given the highly localized nature of the flange. It is possible that the medial flange represents an attachment site for a muscle that was unique to pachycephalosaurs, but this is an unparsimonious assumption from the perspective of the EPB [26]. It is more likely that the flange served as an attachment site for enlarged sacroiliac ligaments, which would have served to brace the pelvis against lateral impacts from head-butting. In most extant tetrapods, the dorsal and ventral sacroiliac ligaments serve to stabilize the sacroiliac joint and pelvis [109]. These ligaments are often a source of injury in animals that experience high intensity or high impact activities involving the hindlegs. Human athletes, for example, will often experience injuries to the sacroiliac joint after an increase in the frequency/intensity of training [110–112]. Similar injuries are often found in work animals experiencing abnormally high pelvic stresses. One such example includes dogs working in the police, military, and search and rescue [109]. These animals are subject to constant high intensity agility training and are often asked to repetitively assume an upright stance which places excessive weight on their hind limbs [109]. As a result, many working dogs experience repeated injuries to the sacroiliac joint and ligaments, which often results in pelvic and lower back pain leading to early retirement [109]. If pachycephalosaurs participated in high impact head-butting competitions, they most certainly experienced high stress levels focused around their pelves and hind limbs to remain upright. If such stresses were a frequent occurrence for these animals, it is likely that they would be at a high risk for sacroiliac joint injuries. Enlarged sacroiliac ligaments may have therefore served to strengthen and stabilize this joint to mitigate pelvic injuries in pachycephalosaurs.

Another hypothesis results from comparison with a possibly analogous structure to the iliac flange in squamates, the iliac tubercle or supracetabular process [113]. This process serves as an attachment area for the iliopubic ligament, iliocostalis, and quadratus lumborum muscles, all of which attach to the dorsal margin and medial surfaces of the process [113]. Squamates that can adopt a facultative bipedal stance display a larger and better developed supracetabular processes, and therefore larger attachment area for these muscles and ligaments [113]. Paparella *et al.* (2020) hypothesize that this not only would have strengthened the affected muscles, but also shifted the center of gravity in these squamates to support stability in bipedal locomotion. Furthermore, the quadratus lumborum inserts on the ventral surfaces of the sacral ribs and functions in stabilizing the axial column [113–116]. If the medial flange of a pachycephalosaur served as an extended attachment area for these same muscles and ligaments, it would have further added to the stability of the spine during head-butting competitions.

2. The broadening of the pelvis and tail base (the latter a result of the wide transverse processes) would have further provided a larger attachment area for the caudofemoralis, ILC, and ISC musculature. Accordingly, these muscles were well-developed in pachycephalosaurs, providing further strength and stability to the hind limbs, pelvis, and tail. The distal placement of the fourth trochanter in *Stegoceras validum* relative to other small ornithischians (e.g., *Parksosaurus* and *Thescelosaurus*) further increased the leverage and forward thrust provided by the caudofemoralis [117].

We thus argue that the varied aberrancies of the pachycephalosaur postcranium largely support the agonistic head-butting hypothesis [13–15, 18], because they all contribute to a strong and stable bipedal stance that would have been beneficial during head-butting competitions. Importantly, the same features numbered above are not typically observed in other small, bipedal ornithischians, so a simple appeal to increased stability during bipedal locomotion does not explain why those features did not evolve convergently in other groups. Our appeal to the head-butting hypothesis [5, 13, 14, 18] explains the existence of these features using a single adaptive scenario, and is therefore parsimonious. The reduction of the pubis in pachycephalosaurs does not obviously fit this model, and is therefore considered in the following section.

### Reduction of the pubis

The reduction of the pubis in pachycephalosaurs (not preserved in UALVP 2, but otherwise known in *Homalocephale calathocercos*, *Pachycephalosaurus sp.*, and an as-yet-undescribed juvenile pachycephalosaur (CMN 22039)) necessitated a rearrangement of the associated musculature to maintain functionality. Most notably, the PIFE 1 and 2 muscles, which function in the flexion and adduction of the hip [81, 83], would have been either lost or relocated to the ischium [44, 63, 90, 91]. Because there is no definitive osteological correlate for PIFE 1 or 2 on the ischium of UALVP 2, it is possible that these muscles were lost in *S. validum*. As a result, their functions would have had to have been compensated for by some other structure. However, these inferences are speculative and cannot be confirmed without further examination of the pubic musculature of pachycephalosaurs. This could be accomplished in future studies by examining the preserved pubis of pachycephalosaurs such as *H. calathocercos* (MPC-D 100/1201) and *Pachycephalosaurus sp.* (ROM 73555) for osteological correlates.

Ankylosaurs show a similar reduction of the pubis to the point where it is completely lost in some taxa [118]. Coombs (1979) [90] noted that the reduced pubis of *Euoplocephalus tutus* would have likely had significant impacts on pelvic musculature. The origin of the AMB, for example, would have shifted from the pubis to the pubic peduncle of the ilium. Additionally,

Coombs (1979) [90] suggested that the PIFE musculature may have had an origin shifted to the proximal ischium or that the muscle was lost completely. Assuming pachycephalosaurs displayed such morphologies is not unreasonable given the extreme reduction of the pubis. However, it is impossible to speculate on this as the specimen lacks a pubis. Although the morphology of the ankylosaur pubis is different than that of pachycephalosaurs, comparing the possible myological implications for a reduced pubis in ankylosaurs to the condition observed in pachycephalosaurs with a preserved pubis (e.g. CMN 22039 and ROM 73555) may help shed light on this strange structure.

## Conclusions

Our study suggests many possible avenues for future research. If the pachycephalosaur skull dome, which grew with positive allometry [119], was used in agonistic combat [13, 18], we might predict that the hind limb muscles used to propel the body forward in such head-butting contests similarly grew with positive allometry. Unfortunately, relevant growth series for the pachycephalosaur postcranium are yet unknown, but the recent identification of CMN 22039 as a juvenile pachycephalosaur [Moore et al., in prep.] may be informative in this regard. The need should also be stressed for dynamic muscle modeling of the pachycephalosaur hind limb [120, 121], which would serve to provide some means of estimating the velocity at which the skull dome could be propelled forward and the impact forces involved. Finally, a reconstruction of the axial musculature would further inform considerations about how the pachycephalosaur postcranium may have been modified to both deliver and receive impact forces associated with the proposed combative behaviours of these animals.

We reconstructed the appendicular myology of *S. validum* by examining the appendicular skeleton of UALVP 2 for osteological correlates and comparing them to the musculature of those taxa comprising the EPB of dinosaurs. The seemingly disparate modifications of the pachycephalosaur postcranium were linked with reference to a common agonistic head-butting hypothesis. We found that the caudofemoralis, iliocaudalis, and ischiocaudalis muscles have large attachment areas resulting in enlarged muscles that acted to strengthen and stabilize the pachycephalosaur pelvis, hind limbs, and tail. Furthermore, the caudofemoralis has a more distal attachment on the femur than other bipedal ornithischians, increasing its leverage and functionality in forward thrust. Distinct striations occur on the dorsal surface of the medial iliac flange and are proposed to be associated with sacroiliac ligaments that served to brace the pelvis. All of these modifications would have been beneficial for agonistic head-butting by strengthening or stabilizing the pelvis and hind limbs.

Our proposed muscle reconstruction is largely limited by a lack of available study material, reflective of the strong preservational bias against the preservation of small skeletons in the fossil record [6, 122, 123]. UAVLP 2 is among the most complete known pachycephalosaur skeletons, although it is missing significant portions of the axial column and distal limb elements. Detailed study of other known postcrania (e.g., *Homalocephale calathocercos*, MPC-D 100/ 1201, *Pachycephalosaurus* sp., ROM 73555), alongside dinosaurs with similar body plans (e.g., *Parksosaurus warrenae* and *Thescelosaurus neglectus*) and analogous morphologies (e.g., ankylosaurs), will serve as a test of our proposed muscle model.

## Supporting information

**S1 Fig. Three-dimensional reconstruction of the pectoral and brachial musculature of *Stegoceras validum*.**
(ZIP)

**S2 Fig. Three-dimensional reconstruction of the pelvic and hind limb musculature of *Stegoceras validum*.**
(ZIP)

## Acknowledgments

We would like to thank Drs. David Evans and Kathreen Ruckstuhl for helpful feedback over the course of this study. We are grateful to Dr. Scott Rufolo and Alan McDonald for providing digital scans of specimens when in person study was impossible due to the COVID-19 pandemic. We would also like to thank Dr. Kevin Seymour and Brian Iwama for allowing access to the collections facilities of the Royal Ontario Museum. Finally, we would like to thank our editor, Dr. Claudia Tambussi, and our reviewers, Drs. Kristyn Voegele, Susannah Maidment, and Andre Rowe, for their constructive feedback during the review process.

## Author Contributions

**Conceptualization:** Bryan R. S. Moore, Jordan C. Mallon.

**Data curation:** Philip J. Currie.

**Funding acquisition:** Jordan C. Mallon.

**Investigation:** Bryan R. S. Moore.

**Methodology:** Bryan R. S. Moore.

**Software:** Mathew J. Roloson.

**Supervision:** R. Timothy Patterson, Jordan C. Mallon.

**Writing – original draft:** Bryan R. S. Moore.

**Writing – review & editing:** Bryan R. S. Moore, Philip J. Currie, Michael J. Ryan, R. Timothy Patterson, Jordan C. Mallon.

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
