## [Decision Letter · Decision Letter 0]

15 Feb 2022

PONE-D-22-00629The appendicular myology of Stegoceras validum (Ornithischia: Pachycephalosauridae) and implications for the head-butting hypothesisPLOS ONE

Dear Dr. Moore,

Thank you for submitting your manuscript to PLOS ONE. After careful consideration, we feel that it has merit but does not fully meet PLOS ONE’s publication criteria as it currently stands. Therefore, we invite you to submit a revised version of the manuscript that addresses the points raised during the review process.

Reviewers make some questions about your work, which is why I have decided to request minor revisions. Basically, I ask you to pay attention to the following criticisms from reviewer 1. The main questions of this reviewer are about the organization in the discussion and conclusions sections.

Reviewers 1 and 2 emphasize the need to increase comparisons as much as possible. If you are not in a position to do so or disagree, please justify it in your response to this editor.

We look forward to receiving your revised manuscript.

Kind regards,

Claudia Patricia Tambussi, Ph.D.

Academic Editor

PLOS ONE

Journal Requirements:

"We would like to thank Drs. David Evans and Kathreen Ruckstuhl for helpful feedback over the course of this study. We are grateful to Dr. Scott Rufolo and Alan McDonald for providing digital scans of specimens when in person study was impossible due to the COVID-19 pandemic. Finally, we would like to thank Dr. Kevin Seymour and Brian Iwama for allowing access to the collections facilities of the Royal Ontario Museum. Funding for this study was provided by a Natural Sciences and Engineering Research Council of Canada Discovery Grant to JCM (RGPIN-2017-06356)."

We note that you have provided funding information. However, funding information should not appear in the Acknowledgments section or other areas of your manuscript. We will only publish funding information present in the Funding Statement section of the online submission form. 

"This study supported by a Natural Sciences and Engineering Research Council of Canada (https://www.nserc-crsng.gc.ca/) Discovery Grant to JCM (RGPIN-2017-06356). The funders had no role in study design, data collection and analysis, decision to publish, or preparation of the manuscript.

This study also supported by a Dinosaur Research Institute (https://www.dinosaurresearch.com) Student Project Grant to BRSM. The funders had no role in study design, data collection and analysis, decision to publish, or preparation of the manuscript."

4. We note that you have referenced (Fechner R. Morphofunctional evolution of the pelvic girdle and hindlimb of Dinosauromorpha on the lineage to Sauropoda. 197 pp. Unpublished Ph. D. dissertation, Fakultät für Geowissenschaften, Ludwig-Maximilians-Universität, Munich. 2009.) which has currently not yet been accepted for publication. Please remove this from your References and amend this to state in the body of your manuscript: (Fechner R. Morphofunctional evolution of the pelvic girdle and hindlimb of Dinosauromorpha on the lineage to Sauropoda. 197 pp. Unpublished Ph. D. dissertation, Fakultät für Geowissenschaften, Ludwig-Maximilians-Universität, Munich. 2009. [Unpublished] as detailed online in our guide for authors

Additional Editor Comments:

Reviewers make some questions about your work, which is why I have decided to request minor revisions. Basically, I ask you to pay attention to the following criticisms from reviewer 1. The main questions of this reviewer are about the organization in the discussion and conclusions sections.

Reviewers 1 and 2 emphasize the need to increase comparisons as much as possible. If you are not in a position to do so or disagree, please justify it in your response to this editor.

Reviewers' comments:

Reviewer's Responses to Questions

**Comments to the Author**

1. Is the manuscript technically sound, and do the data support the conclusions?

Reviewer #1: Yes

Reviewer #2: Yes

Reviewer #3: Yes

2. Has the statistical analysis been performed appropriately and rigorously? 

Reviewer #1: N/A

Reviewer #2: N/A

Reviewer #3: N/A

3. Have the authors made all data underlying the findings in their manuscript fully available?

Reviewer #1: Yes

Reviewer #2: Yes

Reviewer #3: Yes

4. Is the manuscript presented in an intelligible fashion and written in standard English?

Reviewer #1: Yes

Reviewer #2: Yes

Reviewer #3: Yes

5. Review Comments to the Author

Reviewer #1: This manuscript is thorough and detailed. The authors have included numerous muscles, each described in detail. I appreciate that the Results section is clearly written in plain English with citations presented for readers who are looking for the EPB specifics. There is no excessive jargon nor overly-complicated sentence structures. I also like that the authors are looking for functional meaning in the reconstructed myology.

However, I found the Discussion and Conclusion sections difficult to follow. These sections would benefit from further description of (1) the authors’ thought processes and (2) the meaning/significance of the topics chosen for discussion. The Discussion paragraph from lines 1267-1284 presents well-constructed arguments and supporting facts in an easy to follow, logical structure; I therefore believe the writing style of this paragraph would serve as a good template for reformatting the rest of the Discussion. Also, although several of the specific myological arrangements highlighted by the authors in the Discussion could have arisen via selection for intra-specific competition, they would also simply add stability to a biped during locomotion and may have arisen via selection for this ‘benefit’. Such viable alternatives weaken their potential lines of support for inferring head butting behavior in Stegoceras. Thus, my main request of the authors is for them to compare their findings to those from additional other ornithischians and, especially, bipedal nonavian dinosaurs. If the authors can convincingly show that the skeletal and myological features highlighted in their Discussion list are unique specializations seen only in pachycephalosaurs (i.e., they are not also seen in other bipedal nonavian dinosaurs or in ornithischians more broadly), then their conclusions concerning potential head-butting behavior would be better supported.

The authors consistently compare to numerous sauropod and theropod taxa in the Results section, but the addition of consistently comparing to a few ornithischians would go a long way toward strengthening their arguments about the uniqueness of Stegoceras. I recognize not all myological descriptions are conducted in the same format and that it can sometimes be difficult to draw meaningful comparisons, but a plethora of prior publications exist to potentially compare against. The authors already cite a few prior studies, such as those by Dilkes 1999, Langer et al. 2007, and Maidment and Barrett 2011, but they draw few comparisons to the myology of the taxa presented in those studies. Other myology studies the authors could potentially compare against which could strengthen their manuscript might include Fearon and Varricchio 2016 (on Oryctodromeus) and Andrei and Solomon 2013 (on Zalmoxes shquiperorum). The authors do not necessarily need to add more comparisons to all of these taxa, but rather more comparisons in general (against whichever taxa they see as ‘best’/most appropriate) with language to highlight how the myology of S. validum is similar to or differs from the “norm” or “standard” myology of ornithischians as currently understood.

Beyond my two primary concerns summarized above, the following minor concerns are listed below:

At numerous points in the manuscript the authors us the term ‘reptiles’ to discuss a group of organisms in a phylogenetic context. As reptiles are a paraphyletic clade and the context is related to a phylogenetic discussion, it would be better to more specifically list the group/groups the authors are implying. For example: Line 98, Line 698, and Line 1048.

Additionally the authors could improve clarity at a few locations in the Results section:

Hind limb is two words

Lines 345-347 Please make it clear if you are reconstructing with conservative morphology. It seems like it but best to be clear and direct.

Line 406 Does "one" mean they have a tendinous insertion?

Lines 499-500 As written, this is easy to misread or become confused by showing absence. Consider rewriting.

Lines 581-582 Which is "this muscle"? The anconeus or the extensor carpi ulnaris?

Line 598 So is only the origin reconstructed? Unclear.

Lines 716-718 I feel like this statement is correct but inaccurate evolutionary phrasing in the technical sense.

Line 851 What are “these” features?

Line 857 Do you mean the reconstructions are inconsistent or there is disagreement on the EPB attachment locations and divisions? Not clear as written.

Line 1006 Please specify the bone referred to as "its" for clarity.

Lines 1028-1030 Same being that the PIFE 3 in crocs attaches where the ISTR does in birds? Please make clear.

Lines 1342-1344 Better developed in pachycephalosaurs than who?

Also, it should be discussed how stability is more important for any biped than a quadruped and why the morphology of S. validum specifically supports headbutting and not just bipedality. Instances to consider adding/expanding this discussion include:

Lines 33-36 If the pelvic muscles are enlarged to balance reduced pubic muscles, would they still have a benefit to head-butting beyond a balanced condition?

Lines 1333-1335 And added to stability for life as a biped correct?

Lines 1336-1339 Wouldn't this also help in general with being bipedal?

Lines 1359-1365 So your list above summarizes, in general, how more stable Stegocerus is, correct? And then here you are saying that stability is related to or useful for headbutting? Please make sure the text is clear here to help your readers make the connections you are describing.

Other comments:

Line 27 Are the attachments sites enlarge or how did you estimate the size of the muscles?

Line 97 Add (non-avian dinosaurs) after birds for explicit clarity.

Lines 104-111 Are there other specimens of S. validum that preserve postcranium material or not? Is this the best preserved postcranium of all pachycephalsaurs? If these answers are what I suspect they might be, they will add importance to your manuscript.

Lines 118-123 Many of your muscles are missing osteological correlates or the bones to insert on. What do you do about them in blender?

Lines 135-142 Move this paragraph to the methods.

Line 200 Are there osteological correlates on the cervical or dorsal vertebrae or is this reconstruction a prime level of inference? It should be stated either way.

Lines 271-272 If it is some myological reconstructions then Jasinoski citation needs and e.g. but if it is only Jasinoski, then drop the “some”.

Lines 328-329 Saying it is found in both and that it is unequivocal is redundant.

Lines 393-394 Needs to be more like "Though lack of an osteological correlate does not rule out attachment of the SC at the subacromial depression, we reconstruct this muscle as attaching to the scapula along the scapulocoracoid suture ..."

Lines 495-498 Why do the origin and insertion have to move together? Why does a crocodilian origin imply a crocodilian insertion? Please cite literature supporting this idea or instances where origins and insertions have migrated together.

Lines 637-639 Please add something about how there is a general rugose area at this location in S. validum and that makes an independent osteological correlate indistinguishable.

Lines 662-664 Since the bones aren't preserved you don't know where the insertion was and are predicting/hypothesizing where it would have been. Please change the language here to reflect that.

Lines 727-728 Are osteological correlates present?

Lines 742-743 Rewrite this sentence. I agree with the logic but as written it implies that either another muscle was used or not as much flexion of the digits occurs in birds rather than that less muscle force/mucle body was needed to produce the same motion due to the reduction of digits.

Lines 773-780 This reads like discussion

Lines 903-910 I feel like this description of this unknown muscle scar should be its own paragraph not intermixed with a muscle you are not assigning there unless you are trying to say it is an extension of the ILT, but if so this is not clear as written.

Lines 992-1002 If the foot was planted the caudofemoralis would "wag" the tail.

Lines 1021-1024 Please expand this discussion of the differences in attachment sites to clarify how S. validum is similar to Piechowski and Talanda yet different from Maidment and Barrett. It is difficult to follow as written.

Line 1027 Please rewrite so it is clear where the ISTR is in birds. The lateral surface or specifically lateral surface of the shaft.

Lines 1223-1225 This paragraph belongs in the methods

Line 1248 Would “conservative methodologies” or something similar be a better phrase than “caveats used”?

Lines 1248-1250 Does “muscles predicted by the EPB)” and “null hypothesis” mean unequivocal muscles?

Lines 1257-1259 I don't understand how this sentence fits in with the rest of the paragraph.

List that starts on Line 1290 I suggest considering splitting this into two lists. One that is anatomical features not related to myology and one that lists features related to myology. I think more space should be devoted to the second list than the first list as it is directly relevant to your data.

Lines 1318-1320 Is it possible that the transition from quadrapedality to bipedality could cause an increase in stress and result in this morphology?

Line 1360 It isn't really that they need to be explained but that they support that hypothesis.

Lines 1372-1375 Lack of evidence is not evidence so it is possible not definitive.

Lines 1407-1419 Paragraph 2 of the conclusion is all new information which traditionally gets included before the conclusion and the conclusion is more of a summary of the work. Your conclusion could benefit from more of a summary.

Table 1

- You have room to add the abbreviation here too under muscle name. This would be helpful to readers.

- DSC Insertion missing Level of Inference

- SC Origin Levels of Inference flipped for the 2 insertions (always on coracoid, sometimes on the scapula)

- BB Insertion should include the equivocal ulna insertion

- BR Insertion should include the equivocal ulna insertion

Table 2

- EDL should be later distal condyle of femur or anterior cnemial crest, not and

- EDB only shows up in this table, nowhere else in the manuscript

- FDL origin does not match the text description

- TA Insertion missing Level of Inference

- ADF2 missing from table

Figure 1

- Reconstructions are C and D. Scans are A and B

Figure 2

- Reconstructions are C and D. Scans are A and B

Figure 3

- Reconstructions are C and D. Scans are A and B

- If you are splitting the supracoracoideus here, then split it into brevis and longus in Table 1.

Figure 4

- Reconstructions are C and D. Scans are A and B

Figure 5

- Reconstructions are C and D. Scans are A and B

Figure 6

- Reconstructions are C and D. Scans are A and B

Figure 7

- Reconstructions are C and D. Scans are A and B

Figure 8

- Reconstructions are C and D. Scans are A and B

- Part D missing the “e” in “ILFE”

- ED longus or brevis? If ED is meant to represent both than make the name plural in the caption.

Figure 9

- Reconstructions are C and D. Scans are A and B

- ED longus or brevis? If ED is meant to represent both than make the name plural in the caption.

- FMT lateralis or medialis? If FMT is meant to represent both than make the name plural in the caption.

Figure 10

- Reconstructions are C and D. Scans are A and B

- FB in Table 2 and Figure 13. Here it is FBB and FBL which are briefly talked about in the text. Make consistent across manuscript

Figure 11

- Reconstructions are C and D. Scans are A and B

- GSC lateralis or medialis? If GSC is meant to represent both than make the name plural in the caption.

Figure 12

- Specify this is a reconstruction of the pelvic and hind limb musculature

- Why are fewer muscles reconstructed here than in the individual bone figures?

- FDL superficialis or profundus? If FDL is meant to represent both than make the name plural in the caption.

Figure 13

- Specify this is a reconstruction of the pelvic and hind limb musculature

- Why are fewer muscles reconstructed here than in the individual bone figures?

- ED longus or brevis? If ED is meant to represent both than make the name plural in the caption.

- GSC lateralis or medialis? If GSC is meant to represent both than make the name plural in the caption.

- FMT lateralis or medialis? If FMT is meant to represent both than make the name plural in the caption.

Reviewer #2: This is an extremely well-written, well-organised and coherent paper. I have only the most minor of comments. Firstly, for reasons nobody knows, hind limbs is two words, while forelimbs is one word. Please correct throughout! On the review PDF the figures were low resolution and the labels unreadable. I suspect this is a conversion problem, but I would urge you to unbold any labels and check proofs when you get them very carefully. I've previously found figure reproduction in PLoS One to be pretty poor and low resolution, so do check. I have a couple of very very minor comments on the attached. My only other comment, and I hate doing this, because it is obviously the worst kind of review, is that I'm feeling rather hard done by that you haven't cited my 2012 Proc B paper at all. In it (mostly in the supplement), I provided full pectoral and pelvic muscle reconstructions of all major groups of ornithischian dinosaurs, including discussion about how changes in features such as the deltopectoral crest and acromial process affect lines of action and thus function of muscles. You haven't cited that paper AT ALL. I think it is directly relevant to your discussion, as well as to various places where you talk about what previous work has inferred and why. I realise it was in the supplement to the main paper, but I don't think you can ignore it - and the findings of the main paper are relevant too. Please have a look at this paper because while I don't think it'll change any conclusions, I do think it may strengthen arguments - at it's really the only other work on ornithischians.

Susannah Maidment

susannah.maidment@nhm.ac.uk

Reviewer #3: The science is sound throughout the manuscript and it is quite straightforward to read. Citations supporting muscle reconstructions and the methodology are supplied in each section and the limitations of the work are properly addressed in the beginning and end. Additionally, implications beyond head-butting (e.g., carnivory) are addressed which is wonderful to see.

Attached is a word document containing some minor revisions on a line-by-line basis. Most of these pertain to formatting/references, and so nothing that will be majorly time-consuming. Though not required by any means, I do wonder if the authors will consider uploading 3D files of bones/muscles they used onto an online database.

In any case, this was a fascinating work and I am eager to see follow-up research on other pachycephalosaur skeletal material in the future.

6. PLOS authors have the option to publish the peer review history of their article (what does this mean?). If published, this will include your full peer review and any attached files.

Reviewer #1: **Yes: **Kristyn K. Voegele

Reviewer #2: **Yes: **Dr Susannah Maidment

Reviewer #3: **Yes: **Andre J. Rowe

---

## [Author Response · Author response to Decision Letter 0]

14 Apr 2022

Rebuttal Letter

We thank the academic editor and the three reviewers for their comments on our manuscript. Each point raised by the editor and reviewers is addressed below. We hope that our responses to these comments address the concerns of all reviewers and that the manuscript will now be suitable for publication.

Sincerely,

On behalf of all authors,

Bryan Moore

Academic Editor:

 All files and references have been edited where necessary to meet PLOS ONE’s style requirements.

"We would like to thank Drs. David Evans and Kathreen Ruckstuhl for helpful feedback over the course of this study. We are grateful to Dr. Scott Rufolo and Alan McDonald for providing digital scans of specimens when in person study was impossible due to the COVID-19 pandemic. Finally, we would like to thank Dr. Kevin Seymour and Brian Iwama for allowing access to the collections facilities of the Royal Ontario Museum. Funding for this study was provided by a Natural Sciences and Engineering Research Council of Canada Discovery Grant to JCM (RGPIN-2017-06356)."

We note that you have provided funding information. However, funding information should not appear in the Acknowledgments section or other areas of your manuscript. We will only publish funding information present in the Funding Statement section of the online submission form. 

Please remove any funding-related text from the manuscript and let us know how you would like to update your Funding Statement.

 All funding-related text has been removed from the manuscript and added to the Funding Statement section of the online submission form.

3. In your Data Availability statement, you have not specified where the minimal data set underlying the results described in your manuscript can be found. PLOS defines a study's minimal data set as the underlying data used to reach the conclusions drawn in the manuscript and any additional data required to replicate the reported study findings in their entirety. All PLOS journals require that the minimal data set be made fully available.

 We have uploaded our minimum underlying dataset as Supporting Information files upon resubmission. These files include two 3D models of the forelimb and hindlimb musculature of Stegoceras validum.

4. We note that you have referenced (Fechner R. Morphofunctional evolution of the pelvic girdle and hindlimb of Dinosauromorpha on the lineage to Sauropoda. 197 pp. Unpublished Ph. D. dissertation, Fakultät für Geowissenschaften, Ludwig-Maximilians-Universität, Munich. 2009.) which has currently not yet been accepted for publication. Please remove this from your References and amend this to state in the body of your manuscript: (Fechner R. Morphofunctional evolution of the pelvic girdle and hindlimb of Dinosauromorpha on the lineage to Sauropoda. 197 pp. Unpublished Ph. D. dissertation, Fakultät für Geowissenschaften, Ludwig-Maximilians-Universität, Munich. 2009. [Unpublished] as detailed online in our guide for authors.

 All unpublished material has been removed from our references. Wherever the material was referenced in-text has also been replaced by appropriate published material.

 I have reviewed our reference list and have not come across any retracted material. All unpublished material has been removed from the list as per academic editor comment 4. There were several references that were not cited in-text and were part of an older version of the manuscript. These references have been removed from the current manuscript. Several references were added to the list while addressing the comments of the academic reviewers.

These include:

1. Alexander RM. Principles of animal locomotion. Princeton University Press; 2013 Oct 31.

2. Brusatte SL, Benton MJ, Desojo JB, Langer MC. The higher-level phylogeny of Archosauria (Tetrapoda: Diapsida). Journal of Systematic Palaeontology. 2010 Mar 15;8(1):3-47.

3. Galton PM. Notes on Thescelosaurus, a conservative ornithopod dinosaur from the Upper Cretaceous of North America, with comments on ornithopod classification. Journal of Paleontology. 1974 Sep 1:1048-67.

4. Sereno PC. Basal archosaurs: phylogenetic relationships and functional implications. Journal of Vertebrate Paleontology. 1991 Dec 31;11(S4):1-53.

Several other references have been minorly edited or rewritten to meet PLOS ONE’s style requirements.

Reviewer 1:

1. I found the Discussion and Conclusion sections difficult to follow. These sections would benefit from further description of (1) the authors’ thought processes and (2) the meaning/significance of the topics chosen for discussion. The Discussion paragraph from lines 1267-1284 presents well-constructed arguments and supporting facts in an easy to follow, logical structure; I therefore believe the writing style of this paragraph would serve as a good template for reformatting the rest of the Discussion. Also, although several of the specific myological arrangements highlighted by the authors in the Discussion could have arisen via selection for intra-specific competition, they would also simply add stability to a biped during locomotion and may have arisen via selection for this ‘benefit’. Such viable alternatives weaken their potential lines of support for inferring head butting behavior in Stegoceras. Thus, my main request of the authors is for them to compare their findings to those from additional other ornithischians and, especially, bipedal nonavian dinosaurs. If the authors can convincingly show that the skeletal and myological features highlighted in their Discussion list are unique specializations seen only in pachycephalosaurs (i.e., they are not also seen in other bipedal nonavian dinosaurs or in ornithischians more broadly), then their conclusions concerning potential head-butting behavior would be better supported.

 The discussion and conclusions sections of the manuscript have been partially rewritten and rearranged to address the specific concerns of reviewer 1 that were outlined in their pdf review document. These sections should now be easier to read and follow.

 The specific myological attachments that we believe may have arisen via selection for intraspecific competition are not present in other ornithischian bipeds. There is also no evidence to support that the ancestors of pachycephalosaurs were quadrupedal. We have stated as much in the discussion. We hope this clarifies our stance that the unique morphologies observed in pachycephalosaurs arose as adaptations for intraspecific competition and not as general adaptations for bipedality.

2. The authors consistently compare to numerous sauropod and theropod taxa in the Results section, but the addition of consistently comparing to a few ornithischians would go a long way toward strengthening their arguments about the uniqueness of Stegoceras. I recognize not all myological descriptions are conducted in the same format and that it can sometimes be difficult to draw meaningful comparisons, but a plethora of prior publications exist to potentially compare against. The authors already cite a few prior studies, such as those by Dilkes 1999, Langer et al. 2007, and Maidment and Barrett 2011, but they draw few comparisons to the myology of the taxa presented in those studies. Other myology studies the authors could potentially compare against which could strengthen their manuscript might include Fearon and Varricchio 2016 (on Oryctodromeus) and Andrei and Solomon 2013 (on Zalmoxes shquiperorum). The authors do not necessarily need to add more comparisons to all of these taxa, but rather more comparisons in general (against whichever taxa they see as ‘best’/most appropriate) with language to highlight how the myology of S. validum is similar to or differs from the “norm” or “standard” myology of ornithischians as currently understood.

 More comparisons to various ornithischian taxa have been added throughout the results of the manuscript. Examples include comparisons to Zalmoxes, Oryctodromeus, Kentrosaurus, Stegosaurus, Panoplosaurus, Euoplocephalus, Chasmosaurus, Centrosaurus, Hypsilophodon, Lambeosaurus, etc. We hope these comparisons aid in highlighting both the uniqueness of the musculature of Stegoceras and how it is similar in places to other ornithischian taxa.

3. Hind limb is two words.

 All instances of “hindlimb” have been changed to “hind limb”.

4. Lines 345-347 Please make it clear if you are reconstructing with conservative morphology. It seems like it but best to be clear and direct.

 Added that the reconstructed morphology is conservative to the relevant section.

5. Line 406 Does "one" mean they have a tendinous insertion?

 Yes, I have rewritten the line so that this is clear.

6. Lines 499-500 As written, this is easy to misread or become confused by showing absence. Consider rewriting.

 The section has been rewritten more clearly.

7. Lines 581-582 Which is "this muscle"? The anconeus or the extensor carpi ulnaris?

 “This muscle” meant the extensor carpi ulnaris. I have rewritten the sentence so that it is clear.

8. Line 598 So is only the origin reconstructed? Unclear.

 Yes, the insertions are not reconstructed as the elements they would have attached to are not preserved. I have made it clear that they are not reconstructed.

9. Lines 716-718 I feel like this statement is correct but inaccurate evolutionary phrasing in the technical sense.

 I have rephrased the section. It should now be technically correct.

10. Line 851 What are “these” features?

 “These” features refer to the attachment sites observed in the EPB. I have rewritten the sentence to reflect this.

11. Line 857 Do you mean the reconstructions are inconsistent or there is disagreement on the EPB attachment locations and divisions? Not clear as written.

 The homology of the muscle is debated in the literature, meaning that authors disagree on its embryologic origin, its attachment sites, and what muscles should be equated to each other in crocodilians and birds. I have rewritten the sentence to make this clear.

12. Line 1006 Please specify the bone referred to as "its" for clarity.

 I have now specified that “its” refers to the ischium in the manuscript.

13. Lines 1028-1030 Same being that the PIFE 3 in crocs attaches where the ISTR does in birds? Please make clear.

 I have rewritten the section so that it is clear that the “same location” is the lateral ischial shaft.

14. Lines 1342-1344 Better developed in pachycephalosaurs than who?

Also, it should be discussed how stability is more important for any biped than a quadruped and why the morphology of S. validum specifically supports headbutting and not just bipedality. Instances to consider adding/expanding this discussion include: Lines 33-36 If the pelvic muscles are enlarged to balance reduced pubic muscles, would they still have a benefit to head-butting beyond a balanced condition?

Lines 1333-1335 And added to stability for life as a biped correct?

Lines 1336-1339 Wouldn't this also help in general with being bipedal?

Lines 1359-1365 So your list above summarizes, in general, how more stable Stegoceras is, correct? And then here you are saying that stability is related to or useful for headbutting? Please make sure the text is clear here to help your readers make the connections you are describing.

 There is no evidence to support that the ancestors of pachycephalosaurs were quadrupedal or that they went through a shift from quadrupedality to bipedality. We also note that all unique pachycephalosaur morphologies are not present in other bipedal ornithischians and as such, are not likely adaptations for bipedality. The head-butting hypothesis explains all the noted adaptations and explains why they are only present in pachycephalosaurs (since no other dinosaurs are proposed to have taken part in such competitions).

15. Line 27 Are the attachments sites enlarged or how did you estimate the size of the muscles?

 The attachment sites are enlarged, I have rewritten the sentence to make this clear.

16. Line 97 Add (non-avian dinosaurs) after birds for explicit clarity.

 I have added “(non-avian dinosaurs)” for clarity.

17. Lines 104-111 Are there other specimens of S. validum that preserve postcranium material or not? Is this the best preserved postcranium of all pachycephalsaurs? If these answers are what I suspect they might be, they will add importance to your manuscript.

 UALVP 2 is one of the best-preserved pachycephalosaur postcranial skeletons in the world. There is a Mongolian specimen that is arguably better-preserved/more complete, but UALVP 2 is undoubtably the best-preserved specimen of S. validum and the most complete pachycephalosaur skeleton in Canada. I have made this clear in the manuscript.

18. Lines 118-123 Many of your muscles are missing osteological correlates or the bones to insert on. What do you do about them in blender?

 I have added a sentence to make it clear that the form of muscles with missing osteological correlates is inferred from the EPB where possible.

19. Lines 135-142 Move this paragraph to the methods.

I disagree that this paragraph should be moved to the methods section of the paper. The methods section describes the methodology used to examine the specimen and create the muscle reconstruction. This paragraph introduces the elements and muscles included in the reconstruction and does not discuss any methodology.

20. Line 200 Are there osteological correlates on the cervical or dorsal vertebrae or is this reconstruction a prime level of inference? It should be stated either way.

 There are no definitive osteological correlates for the RH on the vertebrae of UALVP 2. The inference level is now clearly stated (1’).

21. Lines 271-272 If it is some myological reconstructions then Jasinoski citation needs and e.g. but if it is only Jasinoski, then drop the “some”.

 I have deleted “some” and specified that the reconstruction being referred to is that of Jasinoski et al. (2006).

22. Lines 328-329 Saying it is found in both and that it is unequivocal is redundant.

 I deleted the redundant portion of the sentence.

23. Lines 393-394 Needs to be more like "Though lack of an osteological correlate does not rule out attachment of the SC at the subacromial depression, we reconstruct this muscle as attaching to the scapula along the scapulocoracoid suture ..."

 The section has been rewritten to the specifications of the reviewer.

24. Lines 495-498 Why do the origin and insertion have to move together? Why does a crocodilian origin imply a crocodilian insertion? Please cite literature supporting this idea or instances where origins and insertions have migrated together.

 The section has been rewritten so that the reconstruction is based on osteological correlates in UALVP 2 and not the notion that the origin of the muscle would have moved with the insertion. It is also noted that the lack of an osteological correlate on the ulna of UALVP 2 does not necessarily rule out an ulnar insertion for the biceps brachii in S. validum.

25. Lines 637-639 Please add something about how there is a general rugose area at this location in S. validum and that makes an independent osteological correlate indistinguishable.

 I have rewritten this section to make it clear that a large area encompassing the attachment sites of several muscles is generally rugose.

26. Lines 662-664 Since the bones aren't preserved you don't know where the insertion was and are predicting/hypothesizing where it would have been. Please change the language here to reflect that.

 I have rephrased the sentence to make it clear that the proposed attachment sites are a hypothesis based on the EPB.

27. Lines 727-728 Are osteological correlates present?

 Yes, I have added that there is an osteological correlate on the entepicondyle.

28. Lines 742-743 Rewrite this sentence. I agree with the logic but as written it implies that either another muscle was used or not as much flexion of the digits occurs in birds rather than that less muscle force/mucle body was needed to produce the same motion due to the reduction of digits.

 I have rewritten the sentence so that it is clear that less muscle body was required to flex the digits.

29. Lines 773-780 This reads like discussion.

 I have removed the section. Now all conclusions drawn about the ILC have been moved to the discussion.

30. Lines 903-910 I feel like this description of this unknown muscle scar should be its own paragraph not intermixed with a muscle you are not assigning there unless you are trying to say it is an extension of the ILT, but if so this is not clear as written.

 This section is now a stand-alone paragraph.

31. Lines 992-1002 If the foot was planted the caudofemoralis would "wag" the tail.

 I added the additional function of “wagging” the tail to the description of the caudofemoralis.

32. Lines 1021-1024 Please expand this discussion of the differences in attachment sites to clarify how S. validum is similar to Piechowski and Talanda yet different from Maidment and Barrett. It is difficult to follow as written.

 I added a phrase that states the condition of UALVP 2 is more similar to that observed in Silesaurus (Piechowski & Tałanda). This should clear up any confusion.

33. Line 1027 Please rewrite so it is clear where the ISTR is in birds. The lateral surface or specifically lateral surface of the shaft.

 I have rewritten the sentence to make it clear that the ISTR in birds originates from the lateral surface of the ischial shaft.

34. Lines 1223-1225 This paragraph belongs in the methods.

 I have rewritten the paragraph so that it reads better as part of the results section.

35. Line 1248 Would “conservative methodologies” or something similar be a better phrase than “caveats used”?

 I have replaced the wording as per the reviewer’s suggestion.

36. Lines 1248-1250 Does “muscles predicted by the EPB” and “null hypothesis” mean unequivocal muscles?

 I have reworded the appropriate sentences so that it is clear these two phrases refer to unequivocal muscles.

37. Lines 1257-1259 I don't understand how this sentence fits in with the rest of the paragraph.

 I have removed the problematic sentence.

38. List that starts on Line 1290 I suggest considering splitting this into two lists. One that is anatomical features not related to myology and one that lists features related to myology. I think more space should be devoted to the second list than the first list as it is directly relevant to your data.

 I have taken the reviewer’s suggestion and rewritten/reorganized the discussion appropriately.

39. Lines 1318-1320 Is it possible that the transition from quadrapedality to bipedality could cause an increase in stress and result in this morphology?

 Both pachycephalosaurs and dinosaurs more generally are plesiomorphically bipedal. We find it unlikely that the adaptations we attribute to the head-butting hypothesis were adaptations for bipedality.

40. Line 1360 It isn't really that they need to be explained but that they support that hypothesis.

 I have rewritten the sentence accordingly.

41. Lines 1372-1375 Lack of evidence is not evidence so it is possible not definitive.

 I have rewritten the sentence so it states it is possible that PIFE 1 and 2 were lost in S. validum, and not definitive.

42. Lines 1407-1419 Paragraph 2 of the conclusion is all new information which traditionally gets included before the conclusion and the conclusion is more of a summary of the work. Your conclusion could benefit from more of a summary.

 We believe this paragraph should remain in the conclusions section. We do not introduce any new information that would affect our conclusions and we believe the conclusions section is where we should suggest future research directions.

43. Comments involving tables:

- You have room to add the abbreviation here too under muscle name. This would be helpful to readers.

- DSC Insertion missing Level of Inference

- SC Origin Levels of Inference flipped for the 2 insertions (always on coracoid, sometimes on the scapula)

- BB Insertion should include the equivocal ulna insertion

- BR Insertion should include the equivocal ulna insertion

Table 2

- EDL should be later distal condyle of femur or anterior cnemial crest, not and

- EDB only shows up in this table, nowhere else in the manuscript

- FDL origin does not match the text description

- TA Insertion missing Level of Inference

- ADF2 missing from table

 All comments involving tables have been appropriately addressed. Regarding the ulnar insertion of the biceps brachii and brachialis, we only reconstruct the radial insertion of these muscles because there is no osteological correlate present for the ulnar insertion in UALVP 2 and the ulnar insertion is equivocal in the EPB. We therefore leave this insertion out of the table but acknowledge that it may have been present in-text.

44. Comments involving figures:

Figure 1

- Reconstructions are C and D. Scans are A and B

Figure 2

- Reconstructions are C and D. Scans are A and B

Figure 3

- Reconstructions are C and D. Scans are A and B

- If you are splitting the supracoracoideus here, then split it into brevis and longus in Table 1.

Figure 4

- Reconstructions are C and D. Scans are A and B

Figure 5

- Reconstructions are C and D. Scans are A and B

Figure 6

- Reconstructions are C and D. Scans are A and B

Figure 7

- Reconstructions are C and D. Scans are A and B

Figure 8

- Reconstructions are C and D. Scans are A and B

- Part D missing the “e” in “ILFE”

- ED longus or brevis? If ED is meant to represent both than make the name plural in the caption.

Figure 9

- Reconstructions are C and D. Scans are A and B

- ED longus or brevis? If ED is meant to represent both than make the name plural in the caption.

- FMT lateralis or medialis? If FMT is meant to represent both than make the name plural in the caption.

Figure 10

- Reconstructions are C and D. Scans are A and B

- FB in Table 2 and Figure 13. Here it is FBB and FBL which are briefly talked about in the text. Make consistent across manuscript

Figure 11

- Reconstructions are C and D. Scans are A and B

- GSC lateralis or medialis? If GSC is meant to represent both than make the name plural in the caption.

Figure 12

- Specify this is a reconstruction of the pelvic and hind limb musculature

- Why are fewer muscles reconstructed here than in the individual bone figures?

- FDL superficialis or profundus? If FDL is meant to represent both than make the name plural in the caption.

Figure 13

- Specify this is a reconstruction of the pelvic and hind limb musculature

- Why are fewer muscles reconstructed here than in the individual bone figures?

- ED longus or brevis? If ED is meant to represent both than make the name plural in the caption.

- GSC lateralis or medialis? If GSC is meant to represent both than make the name plural in the caption.

- FMT lateralis or medialis? If FMT is meant to represent both than make the name plural in the caption.

 All mistakes involving figures and figure captions have been fixed. Abbreviations have been removed from figure captions as the abbreviations are made clear in the Abbreviations section of the manuscript. Regarding figures 12 and 13, not all of the muscles are displayed because the point of these figures is to display the muscular reconstruction as a whole, and not highlight each individual muscle. To do so would require numerous additions to the figures showing the reconstruction from additional views with more steps removing superficial musculature. We believe it is better to show the overall superficial and deep musculature in these figures and attach the full 3D reconstructions as supplementary files so that anyone who wants to look at the muscles in more detail has the freedom to do so. Figure 12 has also been renumbered to Figure 6 (all following figures have been renumbered accordingly) as we feel it is better to present this figure with the pectoral and forelimb muscle reconstructions.

Reviewer 2:

1. hind limbs is two words, while forelimbs is one word. Please correct throughout.

 The error has been corrected throughout the manuscript.

2. On the review PDF the figures were low resolution and the labels unreadable. I suspect this is a conversion problem, but I would urge you to unbold any labels and check proofs when you get them very carefully. I've previously found figure reproduction in PLoS One to be pretty poor and low resolution, so do check.

 The figures reproduced in the PLOS ONE manuscript file were low resolution. I remember during the submission process they specifically stated low resolution versions of the figures would be sent out with the manuscript to the reviewers. The image files I have attached with the manuscript meet all of PLOS ONE’s figure requirements and are high resolution. The figures should all be high resolution in the publication.

3. My only other comment, and I hate doing this, because it is obviously the worst kind of review, is that I'm feeling rather hard done by that you haven't cited my 2012 Proc B paper at all. In it (mostly in the supplement), I provided full pectoral and pelvic muscle reconstructions of all major groups of ornithischian dinosaurs, including discussion about how changes in features such as the deltopectoral crest and acromial process affect lines of action and thus function of muscles. You haven't cited that paper AT ALL. I think it is directly relevant to your discussion, as well as to various places where you talk about what previous work has inferred and why. I realise it was in the supplement to the main paper, but I don't think you can ignore it - and the findings of the main paper are relevant too. Please have a look at this paper because while I don't think it'll change any conclusions, I do think it may strengthen arguments - at it's really the only other work on ornithischians.

 The reviewer is absolutely correct. I have read their paper and it was extremely relevant to our manuscript. I have added numerous references to their paper throughout the manuscript and rewritten several paragraphs to incorporate comparisons to ornithischian dinosaurs pulled directly from their muscular reconstructions.

Reviewer 3:

1. Line 42: Which associated bones specifically?

 The associated bones have now been specifically listed in the sentence.

2. Line 118: Which version of Inkscape specifically?

 The version of Inkscape is now specified.

3. Line 119: Which version of Blender specifically? Versions of the software vary quite a bit.

 The version of Blender is now specified.

4. Line 296: Shouldn’t Fig 11 be changed to Fig 3, as to reflect text order?

 This was actually supposed to be Fig 1. I have corrected the error.

5. Line 352: Brusatte et al. 2010 needs to be listed in the references.

 I have added the reference to the references list.

6. Line 477: Source needs to be in parentheses.

Line 478: Source needs to be in parentheses.

Line 560: Brown, 1908 must be in parentheses.

Line 755: Curry, Roger, and Forster, 2001 must be in parentheses.

Line 756: Osborn, 1905 must be in parentheses.

 These are not regular citations. We are following the International Code of Zoologic Nomenclature which stipulates that upon first mention of a taxa in a manuscript that the scientific name be given in full with the author and year published – non bracketed if the genus name has not changed and bracketed if it has changed.

7. Line 1355: Change ‘theory’ to hypothesis.

 The word has been changed.

8. Line 1681: There are four different Sereno citations in the text but three listed in the reference list.

 I have added the missing Sereno citation to the references list.

---

## [Editor Report · Decision Letter 1]

25 Apr 2022

The appendicular myology of Stegoceras validum (Ornithischia: Pachycephalosauridae) and implications for the head-butting hypothesis

PONE-D-22-00629R1

Dear Dr. %Bryan Robert Schjerning Moore%,

We’re pleased to inform you that your manuscript has been judged scientifically suitable for publication and will be formally accepted for publication once it meets all outstanding technical requirements.

Kind regards,

Claudia Patricia Tambussi, Ph.D.

Academic Editor

PLOS ONE

Additional Editor Comments (optional):

Thank you for considered the feedback by the three reviewers. The main questions raised by the them, especially reviewer 1, have been satisfactorily considered in this new version of the manuscript.

Please, consider at a later stage in the editorial process: add a reference on lines 161-162; delete the double parentheses in table 1, Latissimus dorsi ((LD); please check that “unparsimonious” be a correct word (line 2637)

---

## [Editor Report · Acceptance letter]

13 May 2022

PONE-D-22-00629R1 

The appendicular myology of *Stegoceras validum* (Ornithischia: Pachycephalosauridae) and implications for the head-butting hypothesis 

Dear Dr. Moore:

I'm pleased to inform you that your manuscript has been deemed suitable for publication in PLOS ONE. Congratulations! Your manuscript is now with our production department. 

Kind regards, 

on behalf of

Dr. Claudia Patricia Tambussi 

Academic Editor

PLOS ONE